# CIRCUIT INSIGHTS: TOWARDS INTERPRETABILITY BEYOND ACTIVATIONS

**Elena Golimblevskaia[1], Aakriti Jain[1], Bruno Puri[1,2], Ammar Ibrahim[1],**
**Wojciech Samek[1,2,3], Sebastian Lapuschkin[1,4]**

[1]Department of Artificial Intelligence, Fraunhofer Heinrich Hertz Institute
[2]Department of Electrical Engineering and Computer Science, Technische Universität Berlin
[3]BIFOLD - Berlin Institute for the Foundations of Learning and Data
[4]Centre of eXplainable Artificial Intelligence, Technological University Dublin
 **corresponding authors:** {`wojciech.samek,sebastian.lapuschkin`}`@hhi.fraunhofer.de`

## ABSTRACT

The fields of explainable AI and mechanistic interpretability aim to uncover the internal structure of neural networks, with circuit discovery as a central tool for understanding model computations. Existing approaches, however, rely on manual inspection and remain limited to toy tasks. Automated interpretability offers scalability by analyzing isolated features and their activations, but it often misses interactions between features and depends strongly on external LLMs and dataset quality. Transcoders have recently made it possible to separate feature attributions into input-dependent and input-invariant components, providing a foundation for more systematic circuit analysis. Building on this, we propose WeightLens[1] and CircuitLens[2] , two complementary methods that go beyond activation-based analysis. WeightLens interprets features directly from their learned weights, removing the need for explainer models or datasets while matching or exceeding the performance of existing methods on context-independent features. CircuitLens captures how feature activations arise from interactions between components, revealing circuit-level dynamics that activation-only approaches cannot identify. Together, these methods increase interpretability robustness and enhance scalable mechanistic analysis of circuits while maintaining efficiency and quality.

## 1 INTRODUCTION

Large language models (LLMs) have seen rapid adoption in recent years, including in sensitive domains such as medical analysis (Singhal et al., 2023). Despite their remarkable capabilities, understanding the internal mechanisms of these models is crucial for safe and reliable deployment, yet our knowledge in this area remains limited (Olah et al., 2018; Lapuschkin et al., 2019; Sharkey et al., 2025). Several methods have emerged in the fields of mechanistic interpretability and explainable AI to understand how models encode and utilize mechanisms that influence outputs (Olah et al., 2020; Achtibat et al., 2024; Dreyer et al., 2025). Much of the existing work focuses on circuit discovery, identifying subgraphs responsible for specific tasks (Conmy et al., 2023). However, these studies are mainly limited to toy tasks, and understanding the roles of individual neurons and attention heads still requires extensive manual analysis (Elhage et al., 2022; Wang et al., 2022; Bricken et al., 2023).

Automated interpretability methods were proposed to address these limitations. Bills et al. (2023) suggested using larger LLMs to analyze activation patterns of MLP neurons and generate natural language descriptions. Although promising, such an approach faces the fundamental challenge of polysemanticity of MLP neurons, making them inherently difficult to interpret. This bottleneck prompted the development of Sparse Autoencoders (SAEs), which decompose activations into monosemantic features (Bricken et al., 2023), advancing interpretability pipelines (Templeton et al., 2024; Paulo

---

[1]github.com/egolimblevskaia/WeightLens
[2]github.com/egolimblevskaia/CircuitLens

et al., 2025a). More recently, transcoders were introduced in Dunefsky et al. (2024) and Ge et al. (2024) as an alternative approach for extracting sparse features. Unlike SAEs, which reconstruct activations, transcoders sparsely approximate entire MLP layers, while maintaining a clear separation between input-dependent and weight-dependent contributions. This architecture enables efficient circuit discovery and provides direct attributions to other features, attention heads and vocabulary.

Despite these advances in sparse feature space construction, automated interpretability remains heavily dependent on explainer LLMs, which shifts the solution to the black box problem to yet another black box LLM, resulting in notable safety risks that may produce unfaithful or unreliable explanations (Lermen et al., 2025). Its effectiveness is influenced by the prompt, fine-tuning strategy, and the dataset used for generating explanations. Furthermore, sparse features can still be challenging to interpret (Puri et al., 2025), as they may activate on highly specific patterns that are not easily captured by analyzing activations alone, or may be polysemantic (Kopf et al., 2025).

In this work, we focus on automated interpretability grounded in model weights and circuit structure. Our contributions consist of the following methods:

- *WeightLens*: a framework for interpreting models using their weights and the weights of their transcoders, eliminating dependence on the underlying dataset and explainer LLMs. Descriptions obtained via WeightLens match or exceed baselines for token-based features.
- *CircuitLens*: a framework for circuit-based analysis of activations, extending interpretability to context-dependent features by (i) isolating input patterns triggering feature activations and (ii) identifying which output tokens are influenced by specific features.

Circuit-based approach uncovers patterns missed by activation-only methods, simplifying the explainer LLM's task and improving robustness on smaller datasets by combining weight-based and circuit-based information. Additionally, it handles polysemanticity through circuit-based clustering and combining their interpretations into unified feature descriptions.

## 2 RELATED WORK

Recently, a series of works focused on building automated interpretability pipelines for LLMs (Choi et al., 2024; Paulo et al., 2025a; Puri et al., 2025; Gur-Arieh et al., 2025). Those approaches follow the framework introduced by Bills et al. (2023), which consists of passing a large dataset through a model, collecting maximally activating samples for each MLP neuron, and then forwarding those samples with their per-token activations to a larger LLM to generate natural language descriptions.

Several studies refine this pipeline by focusing on prompt construction and description evaluation. For instance, Choi et al. (2024) and Puri et al. (2025) examine how factors such as the number of samples and the presentation of token activations affect description quality. Choi et al. (2024) further fine-tune an explainer model to produce descriptions conditioned on a feature's activations. Beyond input-based evidence, Gur-Arieh et al. (2025) incorporate output-side information, analyzing not only what inputs trigger a feature but also how that feature influences the model's logits.

These pipelines are applied to different representational units. Early work focuses on MLP neurons (Bills et al., 2023; Choi et al., 2024), while later studies extend them to SAE features, which are generally more interpretable and often monosemantic (Templeton et al., 2024; Gur-Arieh et al., 2025; Paulo et al., 2025a; Puri et al., 2025).

As an alternative to SAEs, Dunefsky et al. (2024) and Ge et al. (2024) introduce transcoders, a sparse approximation of MLP layers that decomposes attributions into input-dependent and input-invariant components. Variants such as skip-transcoders (Paulo et al., 2025b) and cross-layer transcoders (CLTs) (Ameisen et al., 2025) have also been explored, demonstrating through qualitative and quantitative analysis that transcoders match or exceed the interpretability gained through SAEs. Through case studies, Dunefsky et al. (2024) show that transcoder circuits can be used for interpreting a feature's function, although based mainly on manual analysis.

The interpretability of transcoder weights is studied by Ameisen et al. (2025), who show that while some connections appear meaningful, interference is a major challenge. They propose Target-Weighted Expected Residual Attribution (TWERA), which averages attributions across a dataset. However, they find that TWERA weights often diverge substantially from the raw transcoder weights, making the method sensitive to the distribution of the evaluation dataset.

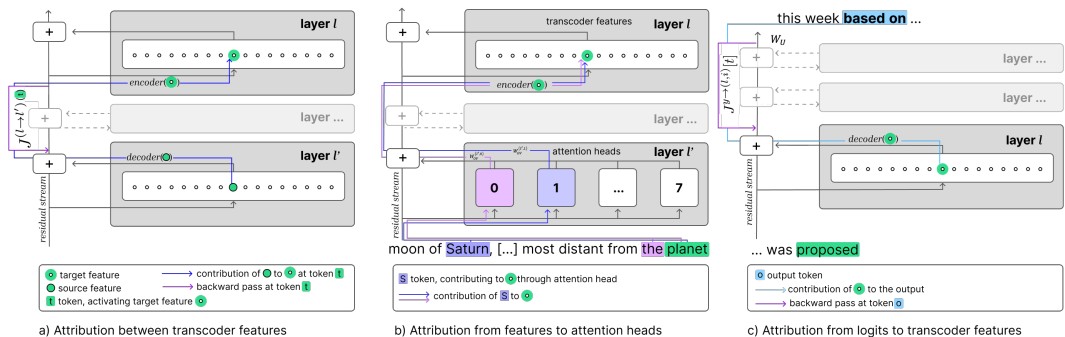

Figure 1: Types of attributions in transcoders.

Finally, Puri et al. (2025) highlight a persistent challenge: even though sparse features, specifically in SAEs, are generally more monosemantic than MLP neurons, they can be highly specific and activate only on certain patterns. This specificity makes them difficult to interpret; either the explainer LLM fails to identify the correct trigger, or the resulting description is too vague to be useful.

## 3 METHODOLOGY

### 3.1 ATTRIBUTION IN TRANSCODERS

We leverage circuit-based information for automated interpretability. Due to their architecture, transcoders enable efficient computation of attributions, which we demonstrate in this section.

As introduced by Dunefsky et al. (2024), given a transcoder structure, the attribution of transcoder feature $i'$ in transcoder layer $l'$ to feature $i$ in layer $l > l'$ on token $t$ can be expressed as:

$$\underbrace{\text{activation}^{(l',i')}[t]}_{\text{input-dependent}} \underbrace{\left( f_{\text{dec}}^{(l',i')} \cdot f_{\text{enc}}^{(l,i)} \right)}_{\text{input-invariant}} \tag{1}$$

where $f_{\text{enc}}^{(l,i)} \in \mathbb{R}^{d_{\text{model}}}$ denotes the $i$-th column of the encoder matrix $W_{\text{enc}}^{(l)} \in \mathbb{R}^{d_{\text{model}} \times d_{\text{features}}}$, and $f_{\text{dec}}^{(l',i')} \in \mathbb{R}^{d_{\text{model}}}$ denotes the $i'$-th row of the decoder matrix $W_{\text{dec}}^{(l')} \in \mathbb{R}^{d_{\text{features}} \times d_{\text{model}}}$, where $d_{\text{features}}$ is the dimension of the transcoder, $d_{\text{model}}$ is the dimension of the model, and $d_{\text{features}} \gg d_{\text{model}}$.

This formulation cleanly separates an input-dependent scalar activation from a fixed, input-invariant connectivity term between features across layers.

Ameisen et al. (2025) propose incorporating a Jacobian term into the attribution formulation, which improves the reliability of feature attributions. With this adjustment, Eq. (1) can be redefined as

$$\text{activation}^{(l',i')}[t] \left( f_{\text{dec}}^{(l',i')} \cdot J^{(l \to l')}[t] \cdot f_{\text{enc}}^{(l,i)} \right), \tag{2}$$

where the Jacobian is given by

$$J^{(l \to l')}[t] := \frac{\partial r_{\text{mid}}^{(l)}[t]}{\partial r_{\text{post}}^{(l')}[t]} \tag{3}$$

with all non-linearities, including attention, normalization, and activation functions, treated as constants with respect to the given input, and $r_{\text{mid}}^{(l)}[t]$ and $r_{\text{post}}^{(l')}[t]$ denote the residual streams before and after the transcoder at layers $l$ and $l'$, respectively (see Figure 1a).

Similarly, we can measure how much previous tokens contributed to the activation of our analyzed feature $(l, i)$ on token $t$ through a specific attention head. This can be done via attribution to that attention head, as presented in Figure 1b (Dunefsky et al., 2024). For an attention head $h$ at layer $l'$ with $l' \leq l$, the contribution of token $s$ through $(l', h)$ to feature $(l, i)$ at token $t$ can be expressed as

$$\underbrace{\text{score}^{(l',h)}(r_{\text{pre}}^{(l')}[t], r_{\text{pre}}^{(l')}[s])}_{\text{attention score from } s \text{ to } t} \underbrace{\left( \left( (W_{\text{OV}}^{(l',h)})^\top f_{\text{enc}}^{(l,i)} \right) \cdot r_{\text{pre}}^{(l')}[s] \right)}_{\text{projection of feature onto head output}}, \tag{4}$$

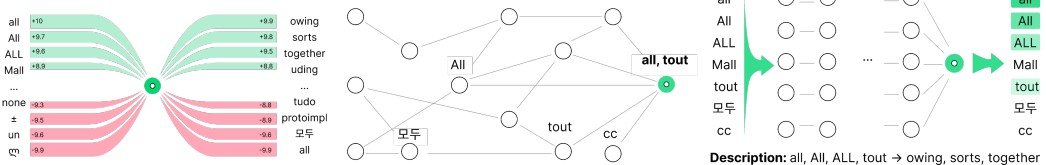

Figure 2: Input-invariant feature analysis with WeightLens. Step 1: Project the feature's encoder and decoder using the embedding and unembedding matrices. Step 2: Identify top contributing features from earlier layers. Step 3: Validate candidate tokens via a forward pass, keeping only those that activate the feature; use them and the output-based outliers for constructing the description.

where $r_{\text{pre}}^{(l')}[s]$ denotes the residual stream at token $s$ before the attention block in layer $l'$, $W_{\text{OV}}^{(l',h)}$ is the output-value matrix of head $h$, and $f_{\text{enc}}^{(l,i)}$ is the encoder vector of feature $(l, i)$.

The contribution of our target feature $(l, i)$ to the output logit $y[t]$ at token $t$ is demonstrated on Figure 1c, and can be expressed as

$$\text{activation}^{(l,i)}[t] \ \left( f_{\text{dec}}^{(l,i)} \ \cdot \ J^{y \rightarrow (l,i)}[t] \ \cdot \ W_U[:, y[t]] \right), \tag{5}$$

where $f_{\text{dec}}^{(l,i)}$ is the decoder vector of feature $(l, i)$, $J^{y \rightarrow (l,i)}[t]$ is the Jacobian from the final residual stream to the post-residual of feature $(l, i)$ at token $t$ (calculated as before with nonlinearities and attention patterns considered as a constant on a given input), and $W_U[:, y[t]]$ is the unembedding vector for token $y[t]$ .

## 3.2 WEIGHTLENS: INPUT-INVARIANT AUTOMATED INTERPRETABILITY

The input-invariant connections from Eq. (1) provide a useful foundation for interpretation of transcoder features, as demonstrated by Dunefsky et al. (2024). Qualitatively, we observe that many strong weight-based connections align with the connections revealed by attribution graphs on feature-relevant inputs, as shown in Appendix E.1. These connections help identify which tokens contribute to a feature, either directly through the embeddings matrix or indirectly by activating upstream features. At the same time, it can be difficult to distinguish meaningful connections that persist when an input is provided to the model, from those that are simply noise. To build on this idea, we make the following assumption:

**Assumption 1:** *Input-invariant connections indicate meaningful structural relationships only if their magnitude significantly exceeds that of other connections, making them statistical outliers.*

Since many features are context-dependent, weight-based analysis alone can be misleading. Connections identified under *Assumption 1* do not always hold across contexts (see Appendix E.3). Since our goal is to identify tokens that reliably characterize a feature, we introduce a validation step to check whether the feature activates on certain tokens in isolation or depends on context, in which case no stable token-level description can be assigned. Formally, we state the assumption:

**Assumption 2:** *If a token is strongly supported by input-invariant connections to the feature (weights) and reflects the feature's inherent behavior, then the feature should activate on this token regardless of context or in the absence of it.*

Based on these assumptions, we generate token-based feature descriptions, where each feature is characterized by tokens that activate it and by tokens it promotes at the output. The procedure is applied layer by layer, progressing from earlier to later layers. The following steps, illustrated in Figure 2, describe the full process.

1. **Embedding and unembedding projections:** Project the feature encoder vector $f_{\text{enc}}$ into the input embedding space using $W_{emb} \cdot f_{\text{enc}}$, and, following Gur-Arieh et al. (2025), the feature decoder vector $f_{\text{dec}}$ into vocabulary logits via $f_{\text{dec}} \cdot W_U$. Detect outliers using $z$-scores (Barnett & Lewis, 1994): embedding-space outliers are candidate tokens for the input-based part of the feature description, and logit-space outliers form the output-based part (outlier threshold 4 for GPT-2 (Radford et al., 2019); 4.5 for Gemma-2-2B (Riviere et al., 2024) and Llama-3.2-1B (Dubey et al., 2024)) .

2. **Analyze weight-based feature connections:** For each earlier layer $l' < l$, compute $W_{\text{dec}}^{(l')} \cdot f_{\text{enc}}^{(l)}$ to identify top contributing features (outliers with $z$-score threshold 3). Inherit their token descriptions as candidate tokens for the input-based part of the feature description.

3. **Validate candidate tokens and generate description:** Keep only tokens that activate the feature in a forward pass. Combine these validated input-based tokens and (optionally) the output-based tokens to form the final feature description. Note that output-based tokens might be noisy and we cannot validate them, therefore this part is optional and its necessity is further evaluated in the Section 4.

Token-based features often respond to multiple forms of the same word. To process the obtained set of tokens and produce a coherent feature description, we apply lemmatization (Bird et al., 2009) to the generated descriptions. This step consolidates different inflected forms into a single canonical form and can be considered a lightweight alternative to LLM-based postprocessing for cleaning and standardizing the descriptions.

### 3.3 CircuitLens: Automated Interpretability Based on Circuits

In transformer models, much of the behavior depends on context and cannot be interpreted from weights alone. Following Bills et al. (2023), most automated interpretability methods select samples that strongly activate a feature across a large dataset and pass them, along with per-token activations, to an explainer LLM to generate a description. However, activations do not always reveal which input tokens caused the feature to fire, and noisy or polysemantic inputs can further reduce description quality. CircuitLens addresses these limitations by providing a circuit-aware approach to feature interpretability. Below, we describe the whole process of automated interpretability with CircuitLens.

**Activation Caching and Sampling** Most prior works (Bills et al., 2023; Choi et al., 2024; Gur-Arieh et al., 2025; Puri et al., 2025) focus on analyzing the most highly activating examples of a feature. This approach is especially sensible for MLP neurons, whose activations are often noisy and unstructured. In contrast, both SAEs and transcoders are explicitly designed to yield monosemantic features. For this reason, we aim to analyze the entire distribution of a feature's activations, in order to capture the broader concept(s) that drive its behavior. For this, we cache maximum sample activations over the whole dataset, which is feasible due to high sparsity of features.

Because activation distributions are typically highly skewed toward zero, we adopt inverse-frequency quantile sampling (De Angeli et al., 2022) to ensure sufficient coverage of rare but strongly activating cases. Specifically, activations are partitioned into $B = 20$ quantile bins. Here, $n_b$ denotes the number of activations in bin $b$, and each activation $i$ in $b$ is assigned weight $w_i = 1/n_b^\alpha$ with $\alpha = 0.9$, and corresponding normalized probability $p_i = w_i / \sum_j w_j$.

Finally, we sample $N = 100$ activations without replacement, producing a diverse set of contexts that up-samples tail cases while still maintaining broad overall coverage.

**Circuit-Based Patterns Detection** A central challenge in interpreting feature activations is that raw activation values do not always reveal what triggered an activation of a given feature. Simply highlighting token activations and prompting language models often yields vague or generic explanations such as "variety of words on variety of topics."[3]

To address this, we focus on identifying *patterns* in the data that both lead to a feature's activation and determine what parts of the output, produced by the model, was influenced by the feature.

- **Input-centric focus**: Using the attribution formulation in Eq. (4), we extract *(attention head, token)* pairs that provide strong contribution for a feature's activation. Outlier pairs are selected based on their $z$-score relative to the distribution of contributions, ensuring that only the strongest connections are retained. We then mask the original input sequence, keeping only tokens that either directly activated the feature or contributed significantly through attention. This procedure isolates interpretable token patterns underlying the activation of a feature, as illustrated in Figure 1b, where the feature activates on references to already mentioned entities.

---

[3]neuronpedia.org/gemma-2-2b/4-gemmascope-transcoder-16k/13598

- **Output-centric analysis**: Using Eq. (5), we evaluate whether the analyzed feature contributed to the prediction of the generated tokens after being activated. This highlights which output tokens were influenced by the feature and thus provides an estimate of its downstream impact, as shown in Figure 1c.

By masking only the input tokens that activated the feature and highlighting only the output tokens it influenced, we pre-identify the relevant patterns ourselves. This removes the burden from the explainer LLM of searching through the full context and instead lets it focus on describing the common concept behind these selected inputs and outputs, hence simplifying its task.

**Circuit-Based Clustering** In order to simplify the explanation task further, we want the explainer LLM to receive clear, monosemantic inputs. However, a single feature can respond to multiple entangled concepts, which remain hard to interpret even in SAEs (Kopf et al., 2025). Semantic clustering of activations in embedding space is not always sufficient, since it ignores the underlying causal, circuit-level mechanisms.

We propose *circuit-based clustering*: for each input, we collect contributing elements, such as transcoder features and token/attention head pairs via Eq. (2) and Eq. (4) respectively. Significant features $(l', i')$ and attention head contributions $(l, h, \Delta)$, where $\Delta$ is the relative token position, are combined into a single vector. Sparse activations ensure that each input has only a few contributors.

To reduce noise, we apply a frequency filter, retaining a feature or head only if it appears in at least a fraction $\rho$ of inputs: $\frac{|\{j:f \in S_j\}|}{N} \geq \rho$, where $S_j$ is the contribution set for input $j$. This step removes features and heads that contribute only for isolated inputs and are unlikely to reflect consistent circuit-level behavior relevant to the feature activation, as well as reduces the size of the set, making subsequent clustering more robust and computationally efficient.

We then compute pairwise Jaccard similarities $J(\mathcal{A}, \mathcal{B}) = |S_{\mathcal{A}} \cap S_{\mathcal{B}}|/|S_{\mathcal{A}} \cup S_{\mathcal{B}}|$, forming an $n \times n$ matrix. Clusters are extracted via DBSCAN (Ester et al., 1996) on this similarity matrix, which is robust to noise and does not require a predefined number of clusters.

**Generating Descriptions** Sampled inputs are first analyzed from input- and/or output-centric perspectives, then grouped into clusters according to the detected circuits. Each cluster is interpreted independently using an explainer LLM (see Appendix F.2 for prompts). Rather than providing full inputs with highlighted activations or token–activation pairs, we supply only the detected pattern, marking the single most activating token. Finally, for each feature, the explainer LLM synthesizes a unified description from the individual cluster-level interpretations.

### 3.4 EVALUATION

All evaluations use the FADE framework (Puri et al., 2025), which relies on an LLM-as-a-judge to assess alignment between a feature and its natural-language description across four metrics: Clarity, Responsiveness, Purity, and Faithfulness. This evaluation setup follows prior work showing strong correlations between such automated judgments and human assessments of description quality.

**Clarity** measures whether the description expresses the concept clearly enough for the explainer LLM to generate synthetic inputs that activate the feature. **Responsiveness** checks whether the feature activates more strongly on these concept-driven inputs than on a randomly sampled dataset. **Purity** measures whether the feature activates primarily on inputs that match the described concept, or also on unrelated inputs. **Faithfulness** evaluates how strongly the feature's activation influences the model's output in the direction implied by the description. A review of automated interpretability metrics, along with details of FADE's computation procedure, is provided in Appendix A.2.

## 4 WEIGHTLENS: EVALUATIONS AND FINDINGS

We analyze the interpretability of transcoders for GPT-2 Small (Dunefsky et al., 2024), Gemma-2-2B (Lieberum et al., 2024), and Llama-3.2-1B (Paulo et al., 2025b), focusing on Gemma-2-2B for qualitative analysis. We evaluate and compare the following setups:

- **WeightLens:** descriptions composed solely of validated tokens that activate the feature.

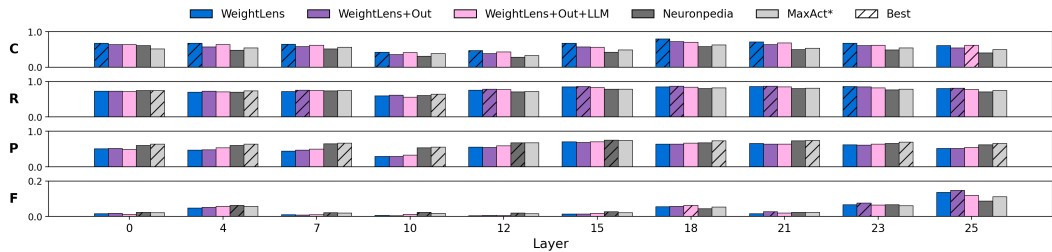

Figure 3: Evaluation of Gemma-2-2B transcoder descriptions (C = Clarity, R = Responsiveness, P = Purity, F = Faithfulness). Methods: WeightLens variants. Baselines: Neuronpedia and MaxAct*.

- **WeightLens+Out:** WeightLens descriptions augmented with tokens promoted by the feature, derived from its unembedding projection.
- **WeightLens+Out+LLM:** descriptions refined using an explainer LLM (gpt-4o-mini-2024-07-18) instead of lemmatization, generating a concise single-line summary based on both activating and promoted tokens (see Appendix F.1 for details).

With these setups we aim to demonstrate the effect of including embedding-based contributions, output-based signals, and LLM refinement on feature description quality. They are further compared to activation-based methods, specifically descriptions available on Neuronpedia (Lin, 2023), as well as generated via MaxAct* method presented by Puri et al. (2025), details are in the Appendix D.

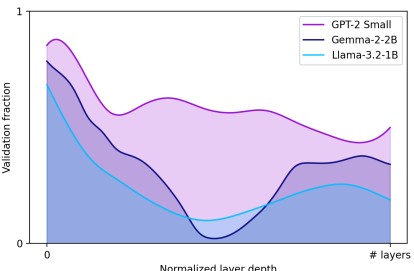

Figure 4: Percentage of validated feature descriptions per layer obtained via WeightLens.

Across ∼250 features per layer, WeightLens methods perform on par with or better than activation maximization methods. They generally achieve higher scores on *Clarity* and *Responsiveness* (Figure 3), while activation-based methods tend to overgeneralize, leading to lower scores. However, activation maximization attains higher *Purity*, highlighting that many features might be context-dependent which is not discovered through WeightLens.

Furthermore, these results demonstrate that although an LLM can produce more general results and filter out noise in the output logits, its utilization is not mandatory in this analysis, since the results are comparable.

Faithfulness is low across layers and methods, with the rise in the later layers, likely due to the transcoder architecture: in contrast to SAEs, which decompose the full residual stream, transcoders write into it like MLPs. Therefore, steering a single feature rarely produces large effects due to model's redundancy. Modifying the faithfulness metric to evaluate interventions on groups of features or on entire circuits rather than individual features might bring more trustworthy evaluations.

Layer-wise analysis of the validated features demonstrates, that token-based interpretability varies strongly with depth (see Appendix G for detailed results). Early layers exhibit clear token-level structure,

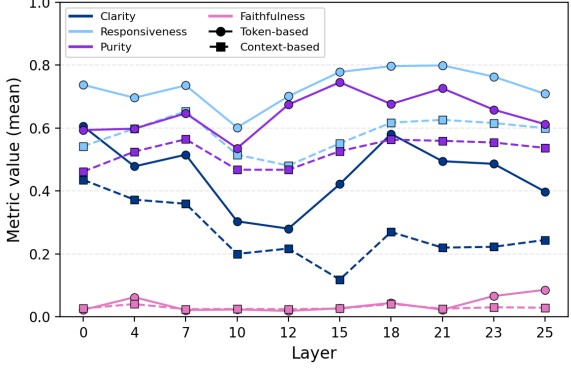

Figure 5: Description quality gap between token-based and context-dependent features based on Neuronpedia descriptions.

making them well-suited for weight-based analysis, with many features activating reliably on specific tokens. In Gemma-2-2B, however, early layers perform slightly worse in terms of descriptions quality than later ones, as demonstrated in Figure 3, reflecting their high activation count ($\ell_0$ of 70–88 for layers 0–7 in contrast to $\ell_0$ of 13–41 for layers 18–25 )[4].

---

[4]huggingface.co/google/gemma-scope-2b-pt-transcoders

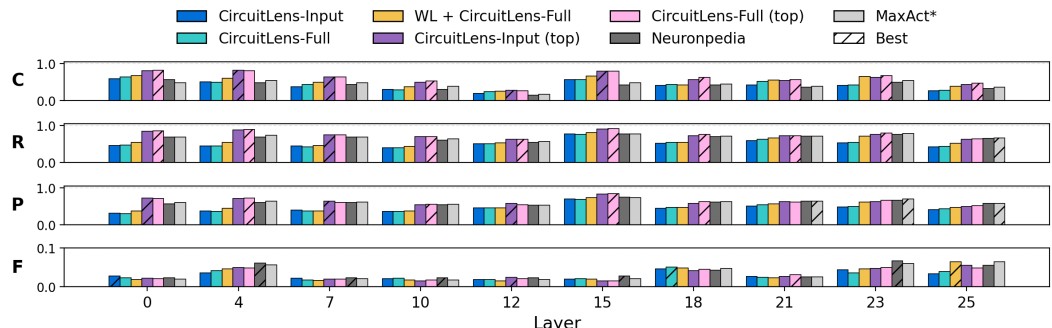

Figure 6: Evaluation of Gemma-2-2B transcoder descriptions (C = Clarity, R = Responsiveness, P = Purity, F = Faithfulness). Methods: CircuitLens variants. Baselines: Neuronpedia and MaxAct*.

Number of token-based features, as well as the quality of their descriptions, drops in the middle layers of Llama and Gemma, as presented in Figure 4, consistent with prior interpretability analysis (Choi et al., 2024), but not in GPT-2, similarity to the results observed by Bills et al. (2023). A likely explanation is the use of RoPE in Llama and Gemma, which introduces additional non-linearities. Within Gemma-2-2B, layer 12 is the least interpretable based on weights: despite high sparsity ($\ell_0 = 6$), it contains few validated token-based features. Majority of the features in this layer are extremely context-dependent and encode very specific patterns.

Later layers partially recover in terms of presence of token-based features, though still below early-layer levels. For instance, Gemma-2-2B layer 21 ($\ell_0 = 13$) shows strong interpretability, with token-based features often acting as key–value pairs that map input tokens to predictable collocations (e.g., *"apologize for"*, *"will be"*).

Additionally, we observe that most interpretable features are token-based. Only a subset of features receive validated input-invariant descriptions: 32.7% for Gemma, 58.8% for GPT-2, and 25.4% for Llama. However, when the weight-based descriptions fail, activation-based ones also perform poorly, as presented in Figure 5.

## 5 CIRCUITLENS: EVALUATIONS AND FINDINGS

For the input-dependent analysis, we evaluate the following approaches, implemented within the CircuitLens framework:

- **CircuitLens-Input:** descriptions derived from the *activation patterns* of the feature, obtained via attribution to attention heads;
- **CircuitLens-Full:** descriptions based on activation patterns obtained through *attributions to attention heads*, augmented with tokens, which generation was influenced by the feature.
- **WeightLens (WL) + CircuitLens-Full:** circuit-based descriptions are *enriched with weight-based tokens* obtained via WeightLens, which are incorporated when merging cluster-level descriptions into a full feature description.

For generating circuit-based descriptions, we use sampling, as described in the Subsection 3.3, on a relatively small dataset of 24M tokens (see Appendix B). In addition, to eliminate the factor of the dataset influence, we compare the circuit-based analysis performed on the same data, as used in MaxAct*, i.e., on a large dataset (2.3B tokens) with sampling from the top, as presented in Appendix B. These results are marked by (top), e.g., CircuitLens-Input (top). For comparison, we use the same MaxAct* and Neuronpedia baselines as in Section 4. Computational requirements for CircuitLens are demonstrated in Appendix C.

Evaluation results, presented in Figure 6, show that circuit-based methods still depend on the dataset, with descriptions from the larger dataset demonstrating the strongest performance across layers, particularly on input-centric metrics. However, descriptions derived from the smaller dataset remain competitive and in some cases outperform activation-based baselines, generated on the larger dataset. Combining weight-based and circuit-based analysis further reduces sensitivity to dataset size and distribution, making interpretability more robust.

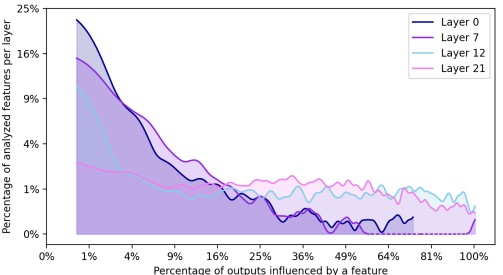

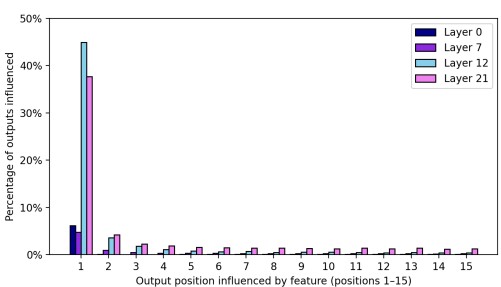

(a) Smoothed histogram of the fraction of features per layer influencing a given percentage of outputs, with both axes square-root transformed.

(b) Fraction of features in each layer that contribute to generated outputs, plotted by output position relative to the activating token.

Figure 7: Influence of features on the new 15 generated tokens from the position of the maximally activating token on a given feature.

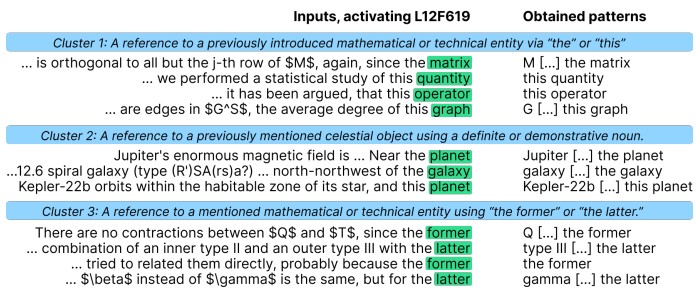

Figure 8: Clusters of activating inputs for L12F619 and their patterns, obtained through attribution to attention heads. Activations are highlighted in green.

Sparse feature descriptions often have low Clarity, as demonstrated by Puri et al. (2025): they fail to specify what precisely triggers a feature's activation. Analysis of the metrics distributions reveals that circuit-based methods yield far fewer features with extremely low clarity compared to purely activation-based approaches (see Appendix G). Additionally, both qualitative and quantitative evidence indicate that sampling from the full distribution, though memory-intensive, provides a more faithful picture of general feature behavior.

We demonstrate in Figure 8 that circuit-based clustering and masking of the input allows uncovering patterns that are not visible based on feature's activation, as illustrated by the feature 619 in layer 12 (L12F619). Within each cluster, some commonality in activating concepts exists, but no clear general pattern emerges from activations alone. Isolating the tokens that contribute most through attention heads reveals that each activating entity is either explicitly mentioned or marked as definite or demonstrative references such as "the" or "this," as well as "former" or "latter," where contributing tokens align semantically.

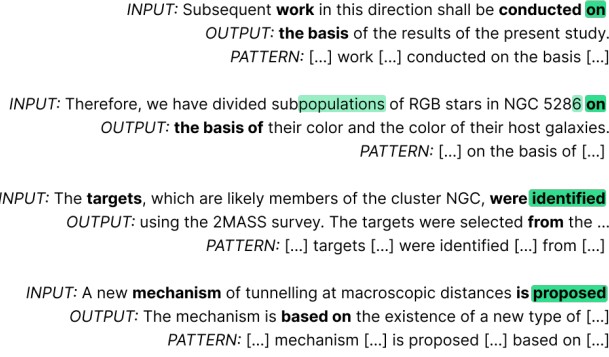

Figure 9: Input- and output-based patterns for feature L21F91 obtained via CircuitLens. Tokens isolated through attribution are shown in **bold**, and activating tokens are highlighted in green.

Extending to the output, the analysis of feature L21F91 demonstrates that functional roles are not fully captured by input activations: while it activates on tokens like "on" or certain verbs, its main effect is generating output phrases such as "the basis of" or "based on" (Figure 9). These contributions, detectable via logit attribution, illustrate how features influence the output, generated by the model.

Despite its importance, output-based analysis is computationally expensive, as each generated token requires both a forward and backward pass. We generate 15 new tokens per sample to assess how many are needed for reliable results. As expected, early layers rarely contribute directly to the output, as shown in Figure 7a, and when they do, the effect is usually limited to the token immediately

following the activating one, see Figure 7b. In deeper layers, most influence remains on the first generated token, though multi-token output patterns also appear, and might be crucial for interpreting a feature's function, particularly in later layers (e.g., feature L21F91 presented in Figure 9).

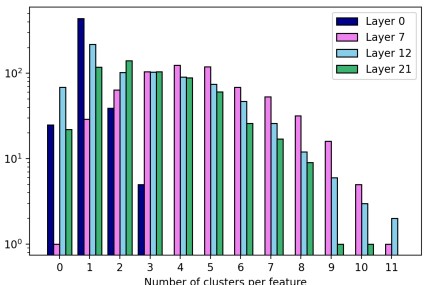

Figure 10: Histogram of number discovered clusters per layer with log scale.

Circuit-based clustering aids in interpreting polysemantic features (Appendix E.4), but its effectiveness varies by layer. Layer 0 shows almost no underlying circuit structure beyond its own attention heads, producing mainly token-based activations, according to Figure 4, and averaging only 1.05 clusters per feature, as demonstrated in Figure 10. Layer 7 exhibits clear polysemanticity, with an average of 4.5 clusters per feature and few single-cluster cases, reflecting extensive circuit formation. Layer 12 is mixed, averaging 2.8 clusters per feature, with many single-cluster or highly clustered features. Qualitative inspection suggests circuit-based clustering captures both main circuits and sub-circuits, and hyperparameter tuning could improve generalization, as can be observed in Figure 8. It also has the largest share of outlier-only features, highlighting challenges in disentangling circuit- from semantic-driven activations. Layer 21 is similar (2.95 clusters/feature) but with fewer extreme cases, consistent with its role in output generation and flexible participation of features in multiple circuits, as visible in Figure 9.

Our results demonstrate that both weight-based and circuit-based interpretability provide meaningful insights, improving precision and scalability in automated feature analysis. For future work, CircuitLens could be extended to architectures beyond transcoders, such as SAEs or crosscoders, by leveraging gradient-based attribution methods. Automated clustering of features could be explored further to interpret larger circuits and efficiently identify and steer functional subcomponents. Additionally, more detailed analysis of clustering hyperparameters would help optimize the balance between capturing polysemantic patterns and avoiding overfragmentation, further enhancing the robustness and applicability of automated interpretability across diverse model families.

## CONCLUSION

In this work, we address a fundamental missing piece in automated interpretability pipelines by developing methods that leverage models' underlying structural information. We show that raw activations alone often fail to reveal the patterns driving feature activation, while transcoders, well-suited for circuit discovery, enable efficient incorporation of structural information. Our proposed frameworks demonstrate that structural information allows more scalable and robust interpretability. The weight-based analysis offers an efficient alternative for context-independent features – covering up to 58.8% of cases – without requiring large datasets or external LLMs. Circuit-based clustering isolates groups of activating texts into more interpretable clusters, while input- and output-based analysis further clarifies each feature's functional role. Together, these methods reduce dependence on large datasets, improve robustness, and make automated interpretability more scalable and practical for real-world applications. By bridging activation-based approaches with weight-based analysis and circuit discovery, our work opens new avenues for understanding model behavior at scale.

## LIMITATIONS

WeightLens enables feature interpretation without relying on a dataset or an explainer LLM, but is specific to the transcoder architecture and cannot be directly applied to SAEs. In addition, context-dependent features are not well captured by this approach. CircuitLens improves robustness to dataset size and distribution, and reduces the complexity of the LLM's task, yet fully eliminating the need for a dataset or an explainer LLM remains challenging. Measuring the downstream effects of features in transcoders is also limited due to high feature redundancy and their architectural differences from residual SAEs that demonstrate good results in steering. As a result, faithfulness scores remain low across all experiments, including the baselines.

ACKNOWLEDGMENTS

We thank Patrick Kahardipraja for valuable discussions and support. This work was in part supported by the European Union's Horizon Europe research and innovation programme (EU Horizon Europe) under grant TEMA (101093003), and by the German Research Foundation (DFG) as research unit DeSBi [KI-FOR 5363] (459422098).

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

# A  EXTENDED RELATED WORK

## A.1  AUTOMATED INTERPRETABILITY

Most work in automated interpretability builds on the pipeline of Bills et al. (2023), where a dataset is passed through GPT-2 (Radford et al., 2019) to collect MLP neuron activations. A larger LLM (GPT-4; (OpenAI andAchiam et al., 2024)) then generates descriptions based on top-k activating sequences, and is further used as an "activations simulator" to evaluate these descriptions.

To address the polysemanticity of MLP neurons, Bricken et al. (2023) propose sparse autoencoders (SAEs). Templeton et al. (2024); Paulo et al. (2025a) extend this pipeline to SAEs and show that their features are more monosemantic and interpretable than MLP neurons.

Subsequent work refines automated interpretability in different ways (Choi et al., 2024; Templeton et al., 2024; Gur-Arieh et al., 2025; Kopf et al., 2025; Paulo et al., 2025a; Puri et al., 2025). Choi et al. (2024) fine-tune a smaller LLM (Llama-3.1-8B-Instruct (Dubey et al., 2024)) for neuron description and evaluation, systematically studying prompt design choices such as token highlighting, token–activation pairs, and number of examples. Puri et al. (2025) run a broader prompt analysis on SAEs, finding results that sometimes contradict Choi et al. (2024), especially on how token activations should be communicated. They also emphasize dependence on explainer model quality and challenges from the fine-grained specificity of SAE features.

Gur-Arieh et al. (2025) combine input- and output-based analysis, showing that descriptions improve when considering both what a feature activates on and what it promotes.

Kopf et al. (2025) address polysemanticity by clustering activating inputs. They sample from the top 1% of activations, embed sequences with gte-Qwen2-1.5B-instruct (Li et al., 2023), and apply k-means clustering into five groups, generating one description per cluster. This consistently outperforms prior work (Bills et al., 2023; Gur-Arieh et al., 2025), demonstrating the benefit of handling polysemanticity directly.

## A.2  EVALUATION METRICS

Developing automated evaluation metrics is essential for interpretability research, since manual assessment of description quality does not scale. Several main approaches have been proposed: (i) simulated activations, where an LLM predicts a feature's activation on text samples given its description (Bills et al., 2023; Choi et al., 2024); (ii) classifier-based metrics, where an LLM judges how strongly a text sample relates to a feature's description (Templeton et al., 2024; Paulo et al., 2025a; Puri et al., 2025); (iii) synthetic data approaches, where an LLM generates or labels data from a description (Gur-Arieh et al., 2025; Puri et al., 2025); and (iv) output-based metrics, which evaluate how much a feature influences model outputs (Bills et al., 2023; Gur-Arieh et al., 2025; Paulo et al., 2025a; Puri et al., 2025).

Simulated-activation metrics (Bills et al., 2023; Choi et al., 2024) are inexpensive but fail to capture many failure modes in description generation (Puri et al., 2025).

Classifier-based metrics instead ask a judge LLM to score how related a sample is to a description, often on a scale from 0 (not related) to 3 (completely related) (Templeton et al., 2024; Puri et al., 2025). Similar detection-based setups appear in Paulo et al. (2025b), where the model identifies which samples match the concept. Evaluation can then be quantified using AUROC scores (Kopf et al., 2025), or metrics such as Gini coefficient and Average Precision, which Puri et al. (2025) combine into Responsiveness and Purity scores.

Synthetic data metrics compare activating and non-activating examples, either LLM-generated or sampled uniformly from a dataset (Gur-Arieh et al., 2025; Puri et al., 2025).

Finally, output-based metrics test whether descriptions capture a feature's causal effect on model outputs. Puri et al. (2025) propose Faithfulness, where a judge LLM rates concept presence in steered generations. Paulo et al. (2025a) introduce Intervention Scoring, while Gur-Arieh et al. (2025) apply a similar approach; where the model's task is to distinguish outputs produced under feature steering from control generations.

In this work, we use the FADE framework to evaluate the obtained descriptions (Puri et al., 2025). It implements the following metrics:

- **Clarity** This metric uses synthetic data generated by an explainer LLM based on the proposed concept description. Feature activations on the synthetic data are compared to a reference distribution of activations on a natural dataset. The Gini coefficient quantifies how well these distributions separate, yielding a score of 1 when the synthetic-data activations are substantially stronger, and 0 when they are indistinguishable. This metric evaluates how clearly the description captures the underlying concept, which is crucial for narrow, fine-grained sparse features.

- **Responsiveness** An LLM-as-a-judge rates how strongly samples from the natural dataset relate to the described concept (on a 0–2 scale). The Gini coefficient then measures whether the feature activates more strongly on concept-related samples than on uniformly drawn natural data, producing a score between 0 and 1. This reflects how responsive the feature is to the described concept.

- **Purity** Using the same data as Responsiveness, this metric replaces the Gini coefficient with Average Precision to quantify whether the feature activates exclusively on the described concept or also on unrelated phenomena. High purity indicates that the description captures all major activation patterns of the feature.

- **Faithfulness** This metric evaluates the downstream influence of the feature. The feature's activation is perturbed, positively scaled, negatively scaled, or fully ablated, and an LLM judges whether the frequency of the described concept in the model's output changes accordingly.

## A.3 CIRCUIT TRACING

A recent advance in mechanistic interpretability is the introduction of transcoders (Dunefsky et al., 2024; Ge et al., 2024). Unlike sparse autoencoders (SAEs), transcoders provide a structured way to trace how upstream feature activations contribute to downstream activations, enabling circuit-level analysis across layers. Their key innovation is the decomposition of a feature's activation into an input-dependent and an input-independent component. The latter depends only on transcoder weights, allowing it to be analyzed separately and efficiently.

Using GPT-2, Dunefsky et al. (2024) demonstrate cases where activating tokens identified through weight-based analysis align with those discovered via traditional activation-based methods. This indicates that transcoders can support both prompt-specific attribution graphs and global, weight-derived connectivity maps.

Extending this work to Gemma-2-2B (Riviere et al., 2024), Ameisen et al. (2025) highlight a major limitation of weight-based analysis: interference from context-dependent components of the architecture. To address this, they introduce *target-weighted expected residual attribution* (TWERA), which adjusts virtual weights using empirical coactivation statistics, effectively up-weighting connections between frequently coactivating features. However, they also show that TWERA can significantly diverge from the original transcoder weights, making it dependent on the dataset used to compute coactivations and limiting its reliability as a fully weight-based method.

## B DATASET PROCESSING

We used the uncopyrighted version of the Pile dataset[5] (Gao et al., 2020) with all copyrighted content removed. This version contains over 345.7 GB of training data from various sources. From this dataset, we extract two datasets of size 6GB (2.3B tokens), and 40MB (24M tokens) for generating MaxAct* descriptions and circuit based descriptions. The extracted portion from the training partition was used to collect the most activated samples based on frequency quantile sampling for the smaller dataset and top percentile sampling for the larger dataset. For evaluations, we utilized the test partition from the same dataset, applying identical preprocessing steps as those used for the training data.

---

[5] huggingface.co/datasets/monology/pile-uncopyrighted

| Stage | # Features | GPU-hours | Description |
|---|---|---|---|
| Activation caching | all layers | 5 | Single forward pass over 24M tokens with sparse activation storage ($\sim$12 GB per layer). This cost is incurred once and shared across all analyses. |
| Input-based feature analysis | 1000 | 8 | Sampling 100 examples per feature and performing forward/backward passes, followed by clustering and bookkeeping. |
| Output-based analysis (5 new tokens) | 1000 | +4 | Additional cost for evaluating next-token effects. Requires one extra forward/backward pass per new token beyond the first. |
| Output-based analysis (15 new tokens) | 1000 | +12 | Estimated cost for broader next-token evaluation, reflecting scaling with the number of additional tokens. |

Table 1: Approximate compute requirements for each stage of the analysis pipeline, measured on an NVIDIA A100 40GB GPU.

Post processing involves several steps to ensure a balanced and informative dataset. First, we used the NLTK (Bird et al., 2009) sentence tokenizer to split large text chunks into individual sentences. We then filtered out sentences in the bottom and top fifth percentiles based on length, as these were typically out-of-distribution cases consisting of single words, characters, or a few outliers. This step helped achieve a more balanced distribution. Additionally, we removed sentences containing only numbers or special characters with no meaningful content. Finally, duplicate sentences were deleted.

## C  COMPUTATIONAL COSTS

To make the computational footprint of our method transparent, we report the approximate runtime measured on a single NVIDIA A100 (40 GB). The activation-caching stage requires a single forward pass over our 24M-token dataset and stores only the sparse activations of Gemmascope transcoders (about 12 GB per layer for 16K-dimensional transcoders). This step takes roughly 5 GPU-hours and is fully amortized across all later analyses. For efficiency, we also allow caching only selected layers. Input-based feature analysis for 1000 features, including sampling, forward and backward passes, clustering, and data-storage operations, requires about 8 GPU-hours. Output-based analysis adds compute depending on the number of next-token effects examined: evaluating a single next token does not introduce additional cost, while evaluating 5 and 15 next tokens adds roughly 4 and 12 GPU-hours respectively. These costs scale predictably with the number of features and enable users to choose an appropriate tradeoff between interpretability depth and available compute.

## D  BASELINES

In this work we use two baselines: Neuronpedia, which provides descriptions available on the Neuronpedia platform (Lin, 2023) and originally generated following Bills et al. (2023), and MaxAct*, introduced in Puri et al. (2025).

To the best of our knowledge, the exact procedure used to generate Neuronpedia descriptions is not documented in detail. Based on Bills et al. (2023), the method relies on selecting the top activating samples from a dataset and highlighting tokens according to their relative activation magnitudes. They use token–activation pairs such as `[This, 0], [feature, 0], [activated, 0], [the, 2], [most, 10]`, where activations are normalized to a fixed range (for example, 0 to 10). Prior work also suggests that using more than 5 to 10 top activating samples provides little improvement.

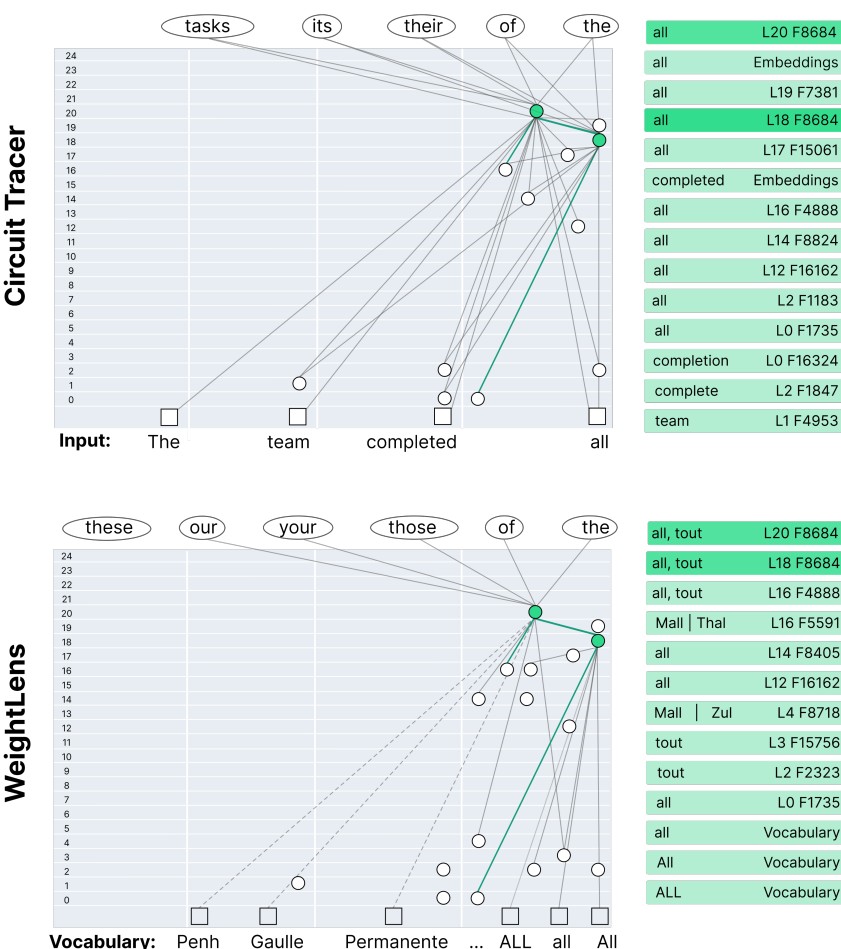

Figure 11: Comparison of attribution graph for features L20F8684 and L18F8684. Top row shows input-dependent graph obtained with Circuit Tracer using the prompt "The team completed all". Bottom row shows weight-based graph for the same features produced by WeightLens.

For MaxAct*, the method caches the top 1000 activations for each feature and uniformly samples 15 inputs from this set. MaxAct* highlights activating tokens using angle brackets, with repeated brackets indicating higher activation strength (see Figure 16 for an example).

# E CASE STUDIES

## E.1 WEIGHTS VS CIRCUITS

A central motivation for weight-based interpretability is the observation that many of the strongest weight connections between features correspond to meaningful computational relationships. Some of these relationships also appear when constructing input-dependent computational graphs with the Circuit Tracer tool. To illustrate this, we examine feature 8684 in layer 20 (L20F8684), which activates on variants of the word "all". Using the Circuit Tracer tool (Hanna et al., 2025) with the prompt "The team completed all"[6] , we compare its input-based circuit to its weight-based connections, as shown in Figure 11.

The input-based circuit contains context-specific edges that depend on the preceding tokens in the prompt. However, two strong upstream connections, to L18F8684 and L16F4888, appear both in WeightLens and in the input-based circuit. While we cannot guarantee that these connections will

---

[6]neuronpedia.org/gemma-2-2b/graph?slug=theteamcompleted

always appear in every context containing the word "all", their presence in both analyses provides supporting evidence for Assumption 1, which states that the strongest weight-based outlier connections tend to generalize across contexts. Examining the connections of L18F8684 further supports this observation. Its link to L0F1735 is visible in the weight-based graph and also appears in the input-based circuit.

Dunefsky et al. (2024) showed that one can use embedding projections to interpret features by identifying which vocabulary tokens contribute directly to a feature's activation. They found that this method works reasonably well for GPT-2. In contrast, our Gemma-2-2B example shows that this technique does not always provide reliable evidence. For L20F8684, the tokens with the strongest weight-based connections ("Penh", "Gaulle", "Permanente") are essentially random and do not reflect the feature's behavior. These tokens also fail to activate the feature when passed through the model. The Circuit Tracer graph provides a clearer explanation: the strongest contributor to L20F8684 is the embedding of "all" (+8.91), and L18F8684 (+8.03) is a close second. This indicates that embedding projections alone are insufficient for identifying tokens that activate a feature.

In contrast, the embedding projection of L18F8684 aligns well with its behavior, showing strong connections to related tokens such as "All", "all", and "ALL". In the input-based circuit the embedding contribution for L18F8684 is +60.67, which is far stronger than any other contributor. The next strongest contributor is L0F1735 (+8.39), which also appears in the weight-based graph.

This example highlights two distinct mechanisms of token-based feature activation. One feature (L18F8684) activates directly when it encounters its preferred token. The other feature (L20F8684) activates when it receives strong upstream input from that first feature, for which we observe extremely strong weight-based connection. WeightLens is designed to leverage exactly these two mechanisms in order to produce consistent token-level interpretations.

## E.2 WEIGHTLENS: FALSE NEGATIVE VALIDATION

As shown above, many connections remain meaningless when we only consider weights. This motivates the validation step proposed in Section 3.2. In this step, we check candidate tokens with a single forward pass to confirm that they truly activate the feature outside of any context. This procedure helps, but it does not always succeed.

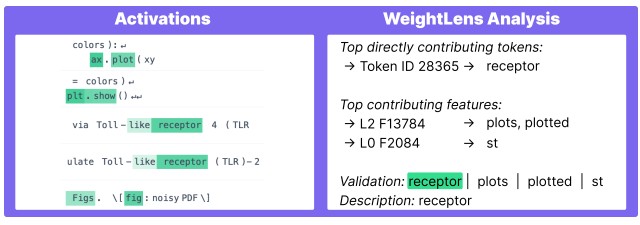

Figure 12: Activations and WeightLens analysis result for L3F9607, with Neuronpedia description "snippets of python code using matplotlib".

Figure 12 illustrates this issue for feature L3F9607[7] . The token "receptor" contributes directly through the embedding matrix, and the single-token forward pass confirms that the feature activates on this input. The broader dataset supports this result, because highly activating samples frequently contain this token.

At the same time, another feature, L2F13784, contributes strongly to L3F9607. It activates on tokens such as "plots" and "plotted", which is intuitively related to our analyzed feature, because L3F9607 fires on plot-related patterns in matplotlib code. However, a single-token forward pass does not activate L3F9607 on these tokens. This tells us that a connection exists in the weights, but we cannot determine whether it is meaningful without input-dependent analysis.

In this case, the method offers only a partial explanation. These limitations directly affect the Purity scores of WeightLens-based descriptions, as discussed in Section 4.

## E.3 WEIGHTLENS: ADDITIONAL EXAMPLES

In Figure 13 we illustrate how the validation step filters out unrelated weight-based connections. The analyzed feature, L21F199, primarily activates on mentions of insects that inhabit plants, in particular larval forms. However, among the strongest weight-based connections we also observe features

---

[7]neuronpedia.org/gemma-2-2b/3-gemmascope-transcoder-16k/9607

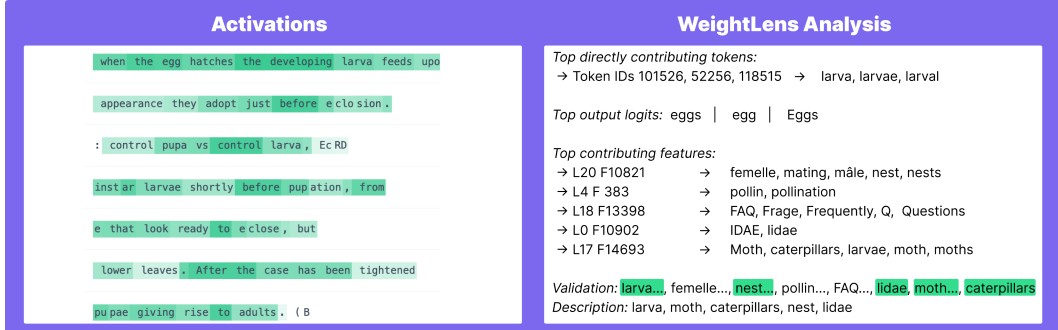

Figure 13: Activations and WeightLens analysis for L21F199 with Neuronpedia description "words related to the life cycle of insects, especially those that live on plants."

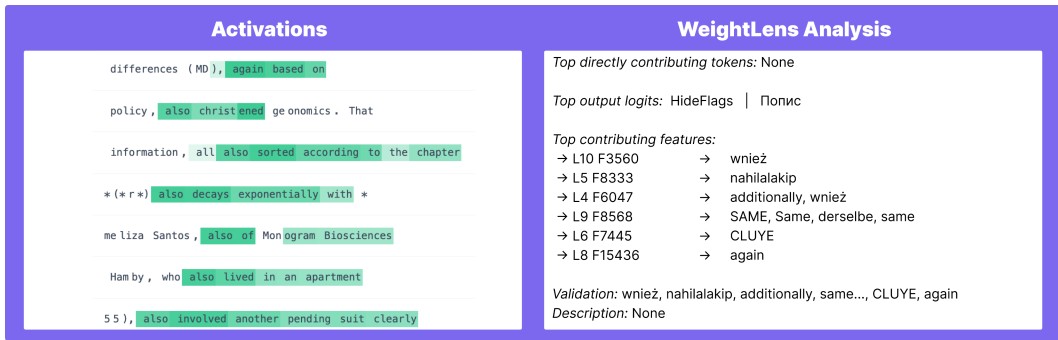

Figure 14: Activations and WeightLens analysis for L12F3146 with Neuronpedia description "words related to describing a process or methodology."

that are unrelated, such as one associated with questions and FAQ-like content. Other connected features contain a mixture of related and unrelated tokens, which makes it difficult to determine which associations truly reflect the behavior of L21F199. The validation step helps disentangle genuine connections from spurious ones.

Middle layers contain many features that activate only in specific contexts, which makes weight-based connections particularly noisy. Although weight-only inspection can occasionally reveal meaningful structure, as seen in Figure 14 for feature L12F3146, it does not provide a principled way to separate signal from noise. Since these intermediate descriptions propagate into later-layer explanations, noise from unvalidated connections would compound and rapidly degrade the quality of feature interpretations.

All pre-computed analyses for every feature of the transcoders are available in our repository, including models such as GPT-2, Gemma-2-2B, and Llama-3.2-1B, allowing readers to explore and reproduce the results in detail.

### E.4    CIRCUITLENS: INTERPRETING PATTERNS AND CLUSTERS

In Figure 16, we illustrate the difference between activation-based description generation and circuit-based description generation. Activation-based methods forward full input sequences to the explainer model, highlighting which tokens activated the feature and by how much, typically relying on the top activations. When a feature exhibits even mild polysemanticity with one dominant mode, these methods almost always recover only that dominant concept. Other activation patterns are frequently absent from the explanation because the dataset distribution over inputs sampled at high activation levels is highly uneven. As a result, which concept becomes represented in the final description largely depends on which activation mode happens to appear among the top examples, rather than reflecting the full behavior of the feature.

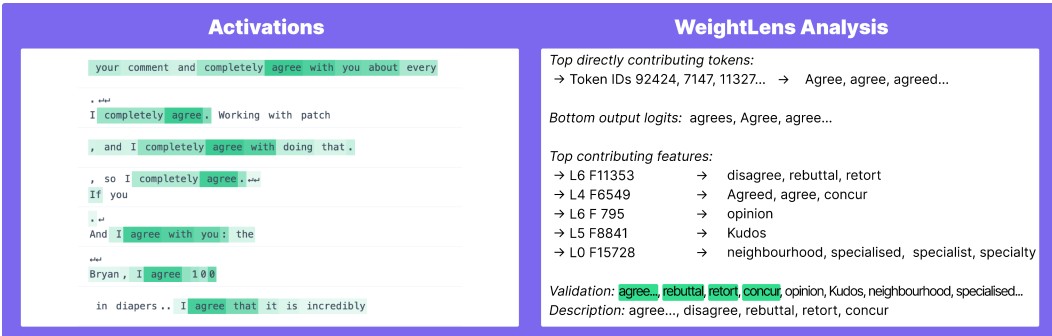

Figure 15: Activations and WeightLens analysis for L7F10119 with Neuronpedia description "instances of agreement and polite conversation."

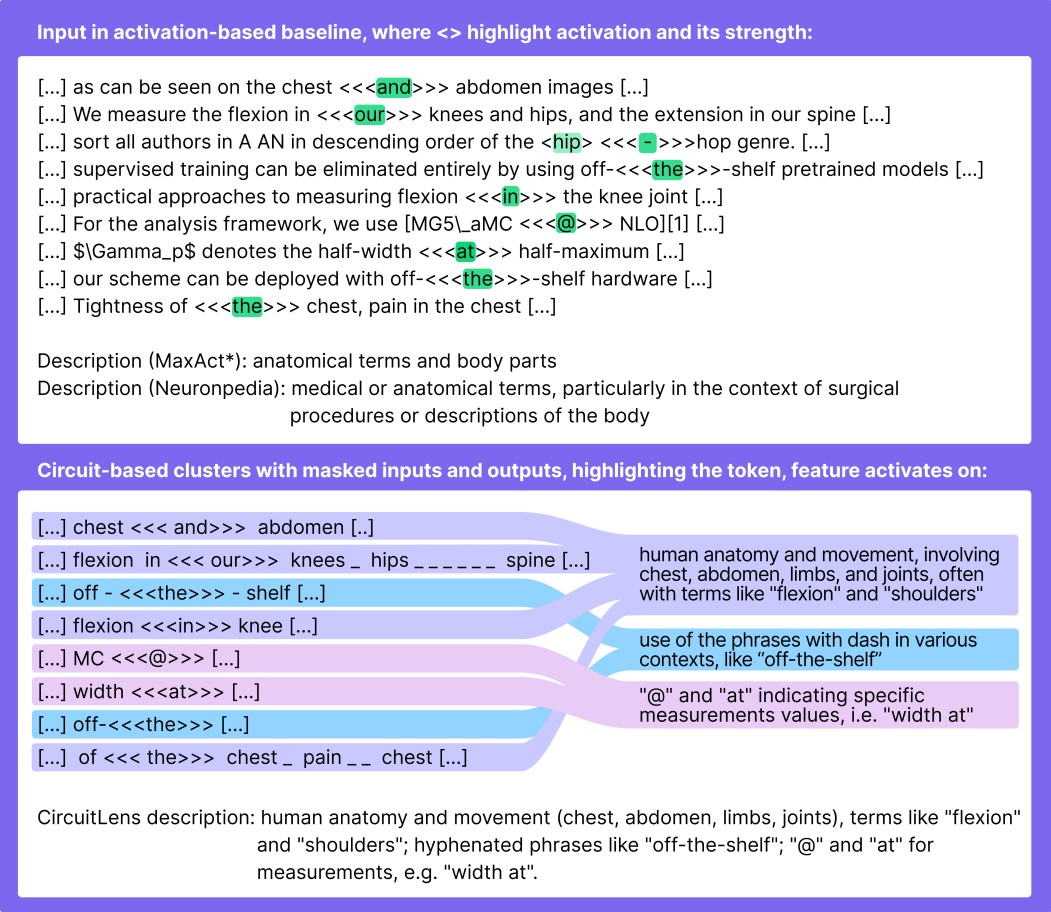

Figure 16: Feature L21F612 (Top) Subset of inputs used to generate descriptions with the MaxAct* baseline; (Bottom) the same inputs masked and clustered with CircuitLens.

To address this, we explicitly sample across the entire activation distribution and subsequently cluster the samples based on the underlying circuits that produced the activation.

A second limitation of activation-based baselines is that a feature may fire on certain words that are not conceptually related to the underlying phenomenon of interest. For instance, in inputs reflecting anatomical contexts in Cluster 1, the feature often activates on high-frequency words such as "and", "our", "in", and "the". This forces the explainer model to infer the relevant concept solely from surrounding context, which is unreliable. In contrast, by masking both the input responsible for

**Input in activation-based baseline, where <> highlight activation and its strength:**

[...] A girl in a knee-length <<<white>>> dress, with a [...]
[...] who was wearing a <<<white>>> labcoat, [...] rubber gloves [...]
[...] so I'm wrapped in a <<<white>>> cotton robe and a pair of black leggings [...]
[...] as she spoke, a child in a <<<blue>>> frock, with a white apron [...]
[...] Before we left for the night, the tables bore <<<green>>> and white balloons [...]
[...] Officer Young examined the bag, a large <<<black>>> plastic bag, and found a large amount [...]
[...] who stood in front of a picture collage of Seath and a cross made of <<<white>>> flowers [...]
[...] Murphy found a stereo speaker and a large <<<green>>> box. [...]
[...] It is the head of a little <<<grey>>> mouse, with a long tail [...]

Description (MaxAct*): description of colors associated with clothing and objects
Description (Neuronpedia): words describing colors or materials

**Circuit-based clusters with masked inputs and outputs, highlighting the token, feature activates on:**

[...] knee-length <<< white>>> dress [..]
[...] <<< white>>> labcoat [...] rubber [...]         white clothes
[...] a <<< white>>> _ robe _ _ _ _ _ leggings [...]
[...] in a <<< blue>>> frock [...]
[...] <<< black>>> plastic bag [...]                  color and material adjectives, often
                                                     describing clothing or containers
[...] of <<< white>>> [...]
[...] <<< green>>> box [...]                          green or black colors applied to
                                                     different containers, i.e. bag, box
[...] <<<grey>>> [...]

CircuitLens description: Color adjectives like "white," "green," and "black" frequently describing clothing,
as well as material adjectives.

Figure 17: (Top) Subset of inputs used to generate descriptions with the MaxAct* baseline; (Bottom) the same inputs masked and clustered with CircuitLens. Pink corresponds to the cluster structure obtained with the clustering hyperparameter `eps = 0.8`. Blue and violet correspond to clusters obtained with `eps = 0.7`. Inputs that are not assigned to either the blue or the violet clusters at `eps = 0.7` are the as outliers under this clustering configuration.

the activation and the model-generated output, we see that the feature in Cluster 1 consistently contributes to producing body-part tokens. Even when no body part has yet appeared in the context, phrases such as "flexion in" lead the model to generate "knee", indicating that we can recover both what triggers the feature and what the feature tends to cause the model to output.

Finally, the cluster-specific circuits reveal that each cluster is supported by a coherent set of upstream features. For example, in Cluster 1 we find features such as L6F2348 (arms anatomy and other body parts), L18F11844 (joints and bone pathologies), and L19F4553 (medical manipulations on animals involving specific body parts). Some features are more general, such as L11F3544 (medicine-related activations), while others capture specific lexical patterns associated with Cluster 1, such as L10F16169 (activations on "our"). In Cluster 2, we observe features including L0F1168 (dash), L6F6265 (tokens following a dash), and L5F6988 (dash-containing expressions such as "by-the-book" or "first-in-first-out"). For Cluster 3, we again find many medicine-related features, as well as features activating on abbreviations (L10F2796, L10F2796) or tokens such as "@" or "at" (L0F2514, L5F14644), even though we have not found monosemantic features corresponding directly to these tokens. This suggests that the broader medical context, rather than specific lexical items, is the primary driver in this cluster.

We also observe several features that appear in the circuits for most inputs, such as L6F2267, L8F3099, and L7F3099. Their activation patterns are dense and difficult to interpret. One possibility is that such features compensate for sparsity or reconstruction-error tradeoffs in the transcoder, although further analysis is required to understand their role.

### E.5 CIRCUITLENS: CLUSTERING HYPERPARAMETERS

In our work, we apply HDBSCAN with `eps = 0.7` and `min_samples = 3`. This configuration frequently yields several narrow clusters, as in Figure 16, while still providing enough inputs within each cluster to support meaningful interpretation. However, such fine grained clustering is not always required. We observe that this parameter setting often subdivides a monosemantic feature into more specific sub-concepts, effectively identifying sub-circuits within the main circuit responsible for the feature's activation. This behavior is illustrated in Figure 17.

The feature in Figure 17 generally activates on colors and occasionally on materials, most prominently when they describe clothing, though not exclusively. A more detailed clustering separates inputs into distinct groups, such as white clothing and green or black boxes or bags. Other instances such as white flowers, a grey mouse, or a blue frock are treated as outliers. In this setting, using a single cluster for all inputs provides a clearer interpretation. Qualitatively, all inputs share circuits containing features related to materials (L2F2256) and to colors (L9F9503, L6F4302, L11F15531, L2F1708, L2F11363, L8F2876, L0F8416, L10F10449, L1F12806). Notably, features L6F2267, L8F3099, and L7F3099 which we previously observed as activating across all inputs also appear consistently in this circuit.

Overall, these observations indicate that circuit-based clustering can reveal both coarse and fine structure in feature behavior, and further work may explore evaluation-guided refinement of clustering parameters and results.

## F    DESCRIPTIONS GENERATION

### F.1    POSTPROCESSING WEIGHT-BASED DESCRIPTIONS

LLM-based postprocessing of weight-based analysis enables generating smoother, more coherent feature descriptions by consolidating input- and output-centric information, specifically, the tokens that activate a feature and those that it promotes or suppresses. This approach is particularly effective at filtering out noise, as promoted or suppressed tokens often include unrelated or random terms that do not reflect the feature's true function.

```
 We're studying neurons in a neural network. Each neuron has certain inputs that activate it
      and outputs that it leads to. You will receive three pieces of information about a neuron
      :

1. The top important tokens.
2. The top tokens it promotes in the output.
3. The tokens it suppresses in the output.

These will be separated into three sections: [Important Tokens], [Text Promoted], and [Text
    Suppressed]. All three are a combination of tokens. You can infer the most likely output
    or function of the neuron based on these tokens. The tokens, especially [Text Promoted]
    and [Text Suppressed], may include noise, such as unrelated terms, symbols, or
    programming jargon. If these are not coherent, you may ignore them. If the [Important
    Tokens] do not form a common theme, you may simply combine the words to form a single
    concept.

Focus on identifying a cohesive theme or concept shared by the most relevant tokens.

Your response should be a concise (1-2 sentence) explanation of the neuron, describing what
    triggers it (input) and what it does once triggered (output). If the input and output are
     related, you may mention this; otherwise, state them separately.

[Concept: <Your interpretation of the neuron, based on the tokens provided>]

Example 1

Input:
[Important Tokens]: ['on', 'pada']
[Tokens Promoted]: ['behalf']
[Tokens Suppressed]: ['on', 'in']

Output:
[Concept: The token "on" in the context of "on behalf of"]
}
...
```

## F.2 GENERATING CIRCUIT-BASED DESCRIPTIONS

At the first step, we treat each cluster of a feature separately. We pass the obtained patterns (input-centric or full, i.e. with patterns detected in the model's output) in there in order to generate a description.

```
You are an explainable AI researcher analyzing feature activations in language models.
You will receive short patterns: fragments of text where tokens activated a feature.
ONLY the snippets shown are the evidence, do not assume any extra surrounding context.

Pattern formatting:
- _ is 1-3 skipped non-important tokens
- [...] is 4 or more skipped not relevant tokens
- The <<<highlighted>>> token of each snippet is usually the most important signal (it is the
    activating token), it can be a part of a word.

Analysis procedure:

1. Do NOT start by interpreting semantics. First treat the data as raw strings.
2. Count and note repeated literal elements (words, single letters, punctuation, suffixes/
    prefixes, LaTeX tokens, different symbols, brackets, arrows, parentheses).
3. Pay special attention to:
  - exact repeated tokens,
  - repeated punctuation or formatting (commas, superscripts, backslashes, braces),
  - positional patterns,
  - capitalization patterns and single-letter variable tokens,
  - functional words, like articles, pronouns, modal verbs, that create a consistent pattern.
4. Only after the literal/structural check, generalize into a short concept (if appropriate).

Decision rules:

- If a single literal token or structural pattern dominates, output that token or structural
    label exactly.
- If it is some grammatical pattern, output exactly that.
- Avoid speculative semantic labels unless literal patterns support them.

Output rules:

- Output exactly ONE concise sentence (<20 words) describing the shared concept or structure.
- If a single token/pattern dominates, output it exactly.
- You may include up to one short example group in parentheses to clarify.
- Do NOT include extra labels or the word "Description:".
- If no clear recurring concept or structure is found, output exactly: No concept found.
- Avoid vague phrases like "in various contexts" or 'a variety of words'.
- Do NOT output your internal reasoning, only the final single sentence.

Example 1

Input:
important to
helps to
permits to
importance to
is possible [...] to
able to
allows us to
purpose [...] is to

Output:
Preposition "to" in phrases that express purpose, intention, or enable an action.
```

At the next iteration, we combine the obtained cluster descriptions into a single one.

```
You are an explainable AI researcher analyzing multiple related concepts.
You will receive a list of **concept descriptions**, each representing a semantic, grammatical
    , or functional element.

[ONLY FOR WeightLens+CircuitLens COMBINED EXPERIMENTS:
Sometimes, you may also receive a phrase at the beginning like "Important tokens: ...", for
    example:
  Important tokens: amazing, largely, upon.
  Important tokens: danger, preparation, prepare, preparing.
  Important tokens: new.
Always integrate these tokens into your description, even if they do not fit naturally with
    the other concept descriptions. ]

Step-by-step reasoning:
```

```
1. Examine all provided concepts carefully. Identify recurring themes, functions, or semantic
     roles.
2. Look for commonalities across the concepts, including:
   - grammatical elements (articles, parts of sentences, syntactic patterns)
   - symbols and punctuation (commas, brackets, etc.)
   - semantic categories
   - mathematical or symbolic markers
3. Pay special attention to specific patterns, which often are described through function
     words (articles, modal verbs, etc.).
4. Merge similar or overlapping elements into a single, concise idea.
5. Think step by step:
   a) Identify the core function or role each concept serves.
   b) Group related concepts together.
   c) Combine them into one coherent description.

Output rules:
- Output exactly **one concise sentence** (<30 words) describing the shared concept or several
     main concepts.
- Include all major elements, but merge overlapping items.
- Include short examples of terms or specific patterns, if they clarify the concept.
- Include any "Important tokens" explicitly in the description.
- Do not add labels, headings, or extra commentary.
- Be precise, avoid speculation, and avoid vague expressions like "in various contexts."
```

# G    EXTENDED RESULTS

In this section, we provide plots for distributions of metrics, presented in Sections 4 and 5. Specifically, Figures 18 and 19 correspond to the results, presented in Figure 3, and Figures 20 and 21 correspond to the scores, demonstrated in Figure 6. In addition, Tables 2 and 3 present corresponding metrics in a tabular form.

Presented results demonstrate that on Clarity our WeightLens and CircuitLens methods, as well as their combination, outperform activation-based baseline. WeightLens results enhance circuit-based information, and allow bridging this gap between smaller and larger datasets, or even outperform the activation-based baseline, thus making the automated interpretability more robust to the dataset imperfections.

| Metric | Method | 0 | 4 | 7 | 10 | 12 | 15 | 18 | 21 | 23 | 25 |
|---|---|---|---|---|---|---|---|---|---|---|---|
| Clarity | WL | 0.67 | 0.67 | 0.65 | 0.42 | 0.47 | 0.67 | 0.80 | 0.71 | 0.67 | 0.61 |
| | WL+Out | 0.63 | 0.57 | 0.58 | 0.36 | 0.38 | 0.57 | 0.72 | 0.63 | 0.60 | 0.54 |
| | WL+Out+LLM | 0.64 | 0.63 | 0.62 | 0.41 | 0.43 | 0.56 | 0.70 | 0.68 | 0.62 | 0.61 |
| | Neuronpedia | 0.61 | 0.48 | 0.52 | 0.30 | 0.28 | 0.42 | 0.58 | 0.50 | 0.49 | 0.40 |
| | MaxAct* | 0.51 | 0.54 | 0.56 | 0.38 | 0.33 | 0.49 | 0.62 | 0.53 | 0.54 | 0.49 |
| Responsiveness | WL | 0.72 | 0.70 | 0.72 | 0.59 | 0.76 | 0.85 | 0.85 | 0.86 | 0.86 | 0.80 |
| | WL+Out | 0.73 | 0.73 | 0.75 | 0.61 | 0.78 | 0.86 | 0.86 | 0.87 | 0.85 | 0.81 |
| | WL+Out+LLM | 0.72 | 0.71 | 0.74 | 0.56 | 0.77 | 0.83 | 0.84 | 0.84 | 0.82 | 0.77 |
| | Neuronpedia | 0.74 | 0.70 | 0.74 | 0.60 | 0.70 | 0.78 | 0.80 | 0.80 | 0.76 | 0.71 |
| | MaxAct* | 0.74 | 0.74 | 0.75 | 0.64 | 0.71 | 0.78 | 0.82 | 0.81 | 0.78 | 0.75 |
| Purity | WL | 0.50 | 0.47 | 0.44 | 0.29 | 0.55 | 0.70 | 0.63 | 0.65 | 0.62 | 0.51 |
| | WL+Out | 0.51 | 0.47 | 0.47 | 0.29 | 0.54 | 0.69 | 0.64 | 0.63 | 0.61 | 0.51 |
| | WL+Out+LLM | 0.49 | 0.53 | 0.49 | 0.33 | 0.59 | 0.70 | 0.66 | 0.63 | 0.63 | 0.54 |
| | Neuronpedia | 0.60 | 0.60 | 0.65 | 0.54 | 0.67 | 0.75 | 0.68 | 0.73 | 0.66 | 0.61 |
| | MaxAct* | 0.64 | 0.63 | 0.66 | 0.55 | 0.67 | 0.73 | 0.73 | 0.74 | 0.70 | 0.65 |
| Faithfulness | WL | 0.02 | 0.05 | 0.01 | 0.01 | 0.01 | 0.01 | 0.06 | 0.02 | 0.07 | 0.14 |
| | WL+Out | 0.02 | 0.05 | 0.01 | 0.00 | 0.01 | 0.01 | 0.06 | 0.03 | 0.07 | 0.15 |
| | WL+Out+LLM | 0.01 | 0.06 | 0.01 | 0.01 | 0.01 | 0.02 | 0.06 | 0.02 | 0.06 | 0.12 |
| | Neuronpedia | 0.02 | 0.06 | 0.02 | 0.02 | 0.02 | 0.03 | 0.04 | 0.02 | 0.07 | 0.09 |
| | MaxAct* | 0.02 | 0.06 | 0.02 | 0.02 | 0.01 | 0.02 | 0.05 | 0.02 | 0.06 | 0.11 |

Table 2: WeightLens-based methods: Mean metric scores across layers

| Metric | Method | 0 | 4 | 7 | 10 | 12 | 15 | 18 | 21 | 23 | 25 |
|--------|--------|---|---|---|----|----|----|----|----|----|----|
| | CL-Input | 0.59 | 0.51 | 0.37 | 0.30 | 0.19 | 0.56 | 0.41 | 0.41 | 0.41 | 0.27 |
| | CL-Full | 0.64 | 0.49 | 0.43 | 0.29 | 0.24 | 0.56 | 0.43 | 0.52 | 0.42 | 0.27 |
| | WL+CL-Full | 0.68 | 0.60 | 0.49 | 0.37 | 0.25 | 0.66 | 0.43 | 0.55 | 0.65 | 0.38 |
| Clarity | CL-Input (top) | 0.80 | 0.81 | 0.64 | 0.50 | 0.27 | 0.80 | 0.57 | 0.54 | 0.63 | 0.43 |
| | CL-Full (top) | 0.82 | 0.81 | 0.64 | 0.53 | 0.27 | 0.79 | 0.62 | 0.56 | 0.68 | 0.46 |
| | Neuronpedia | 0.57 | 0.48 | 0.43 | 0.30 | 0.14 | 0.42 | 0.42 | 0.36 | 0.49 | 0.32 |
| | MaxAct* | 0.49 | 0.55 | 0.48 | 0.38 | 0.17 | 0.49 | 0.44 | 0.39 | 0.54 | 0.36 |
| | CL-Input | 0.46 | 0.45 | 0.45 | 0.40 | 0.51 | 0.77 | 0.53 | 0.59 | 0.54 | 0.43 |
| | CL-Full | 0.47 | 0.45 | 0.43 | 0.40 | 0.51 | 0.76 | 0.54 | 0.64 | 0.55 | 0.44 |
| | WL+CL-Full | 0.55 | 0.55 | 0.46 | 0.44 | 0.53 | 0.81 | 0.54 | 0.67 | 0.72 | 0.52 |
| Responsiveness | CL-Input (top) | 0.85 | 0.88 | 0.75 | 0.71 | 0.63 | 0.91 | 0.72 | 0.72 | 0.77 | 0.63 |
| | CL-Full (top) | 0.86 | 0.89 | 0.75 | 0.71 | 0.63 | 0.92 | 0.76 | 0.73 | 0.80 | 0.64 |
| | Neuronpedia | 0.70 | 0.70 | 0.69 | 0.60 | 0.55 | 0.78 | 0.70 | 0.71 | 0.76 | 0.66 |
| | MaxAct* | 0.69 | 0.74 | 0.70 | 0.64 | 0.57 | 0.78 | 0.71 | 0.71 | 0.78 | 0.66 |
| | CL-Input | 0.31 | 0.37 | 0.40 | 0.35 | 0.46 | 0.70 | 0.44 | 0.51 | 0.48 | 0.41 |
| | CL-Full | 0.30 | 0.36 | 0.38 | 0.36 | 0.46 | 0.69 | 0.47 | 0.54 | 0.49 | 0.43 |
| | WL+CL-Full | 0.38 | 0.44 | 0.37 | 0.38 | 0.46 | 0.74 | 0.47 | 0.56 | 0.61 | 0.47 |
| Purity | CL-Input (top) | 0.73 | 0.71 | 0.64 | 0.54 | 0.57 | 0.83 | 0.58 | 0.63 | 0.63 | 0.50 |
| | CL-Full (top) | 0.71 | 0.72 | 0.61 | 0.55 | 0.54 | 0.84 | 0.62 | 0.62 | 0.66 | 0.52 |
| | Neuronpedia | 0.57 | 0.60 | 0.60 | 0.54 | 0.53 | 0.75 | 0.62 | 0.64 | 0.66 | 0.58 |
| | MaxAct* | 0.60 | 0.64 | 0.61 | 0.55 | 0.53 | 0.73 | 0.62 | 0.64 | 0.69 | 0.58 |
| | CL-Input | 0.03 | 0.04 | 0.02 | 0.02 | 0.02 | 0.02 | 0.05 | 0.03 | 0.04 | 0.03 |
| | CL-Full | 0.02 | 0.04 | 0.02 | 0.02 | 0.02 | 0.02 | 0.05 | 0.02 | 0.04 | 0.04 |
| | WL+CL-Full | 0.02 | 0.05 | 0.02 | 0.02 | 0.02 | 0.02 | 0.05 | 0.02 | 0.05 | 0.06 |
| Faithfulness | CL-Input (top) | 0.02 | 0.05 | 0.02 | 0.02 | 0.02 | 0.02 | 0.04 | 0.03 | 0.05 | 0.06 |
| | CL-Full (top) | 0.02 | 0.05 | 0.02 | 0.02 | 0.02 | 0.02 | 0.05 | 0.03 | 0.05 | 0.05 |
| | Neuronpedia | 0.02 | 0.06 | 0.02 | 0.02 | 0.02 | 0.03 | 0.04 | 0.03 | 0.07 | 0.06 |
| | MaxAct* | 0.02 | 0.06 | 0.02 | 0.02 | 0.02 | 0.02 | 0.05 | 0.03 | 0.06 | 0.06 |

Table 3: CircuitLens-based methods: Mean metric scores across layers

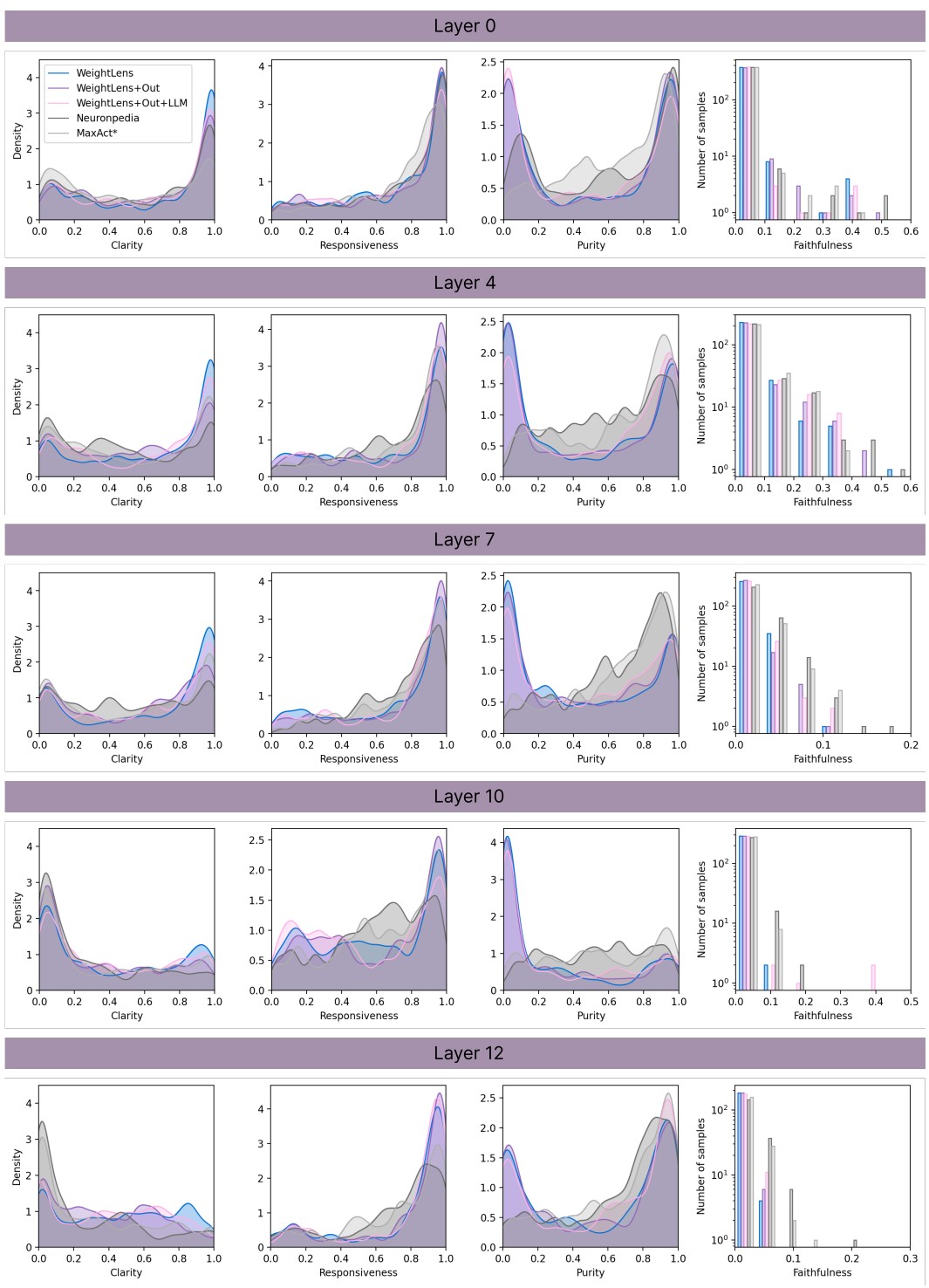

Figure 18: Kernel density estimates illustrating evaluation results of WeightLens methods in comparison to the baselines (layers 0–12).

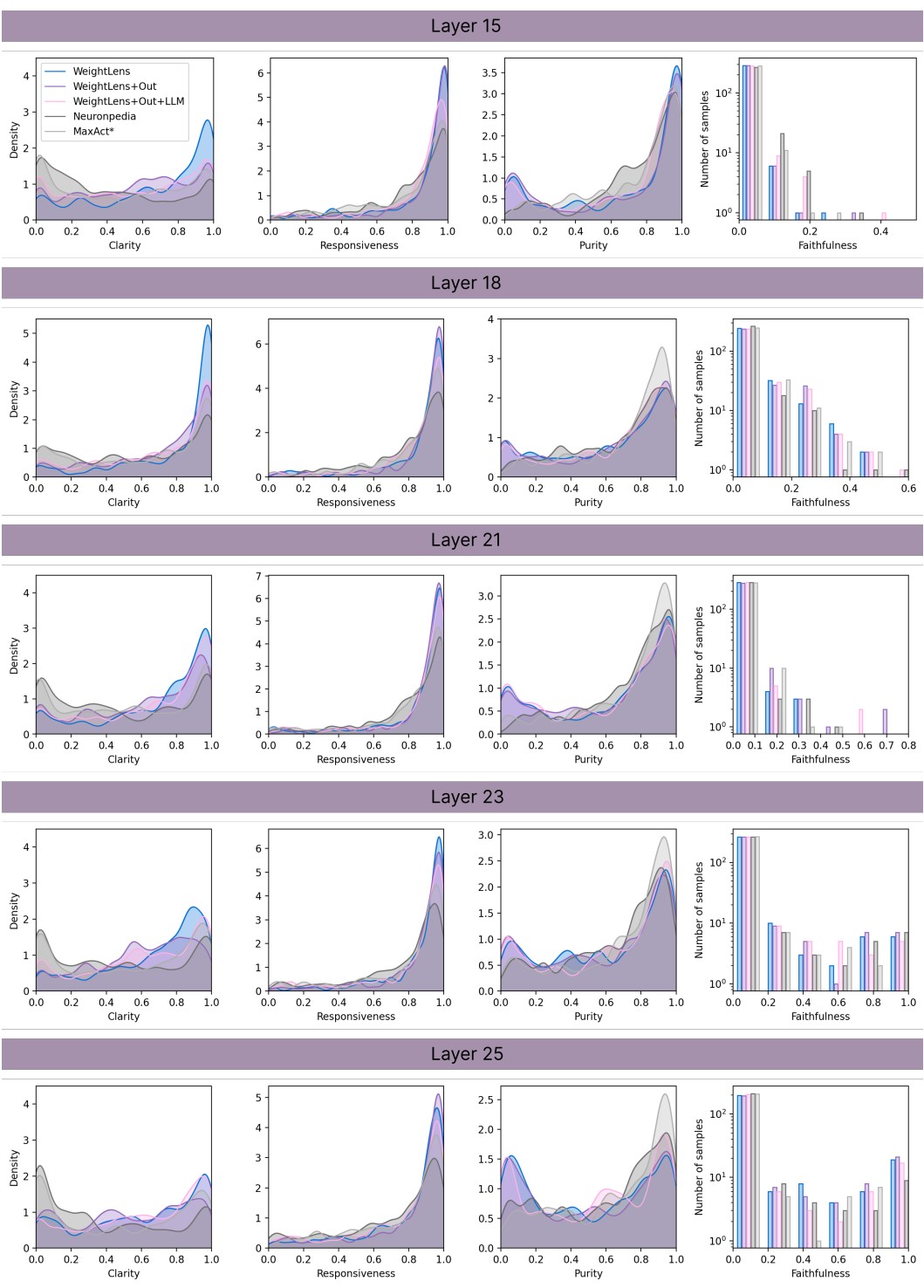

Figure 19: Kernel density estimates illustrating evaluation results of WeightLens methods in comparison to the baselines (layers 15–25).

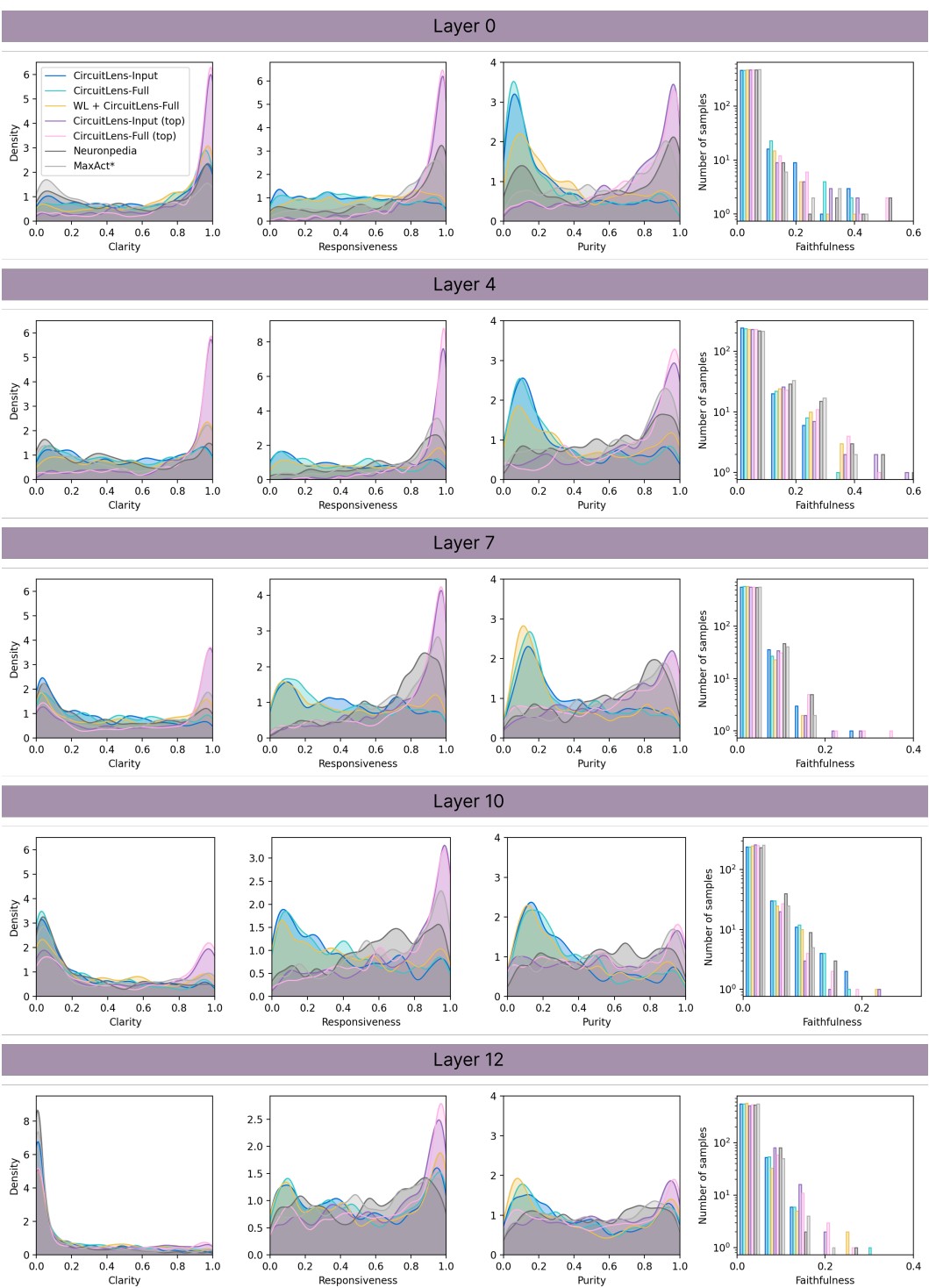

Figure 20: Kernel density estimates illustrating evaluation results of CircuitLens methods in comparison to the baselines (layers 0–12).

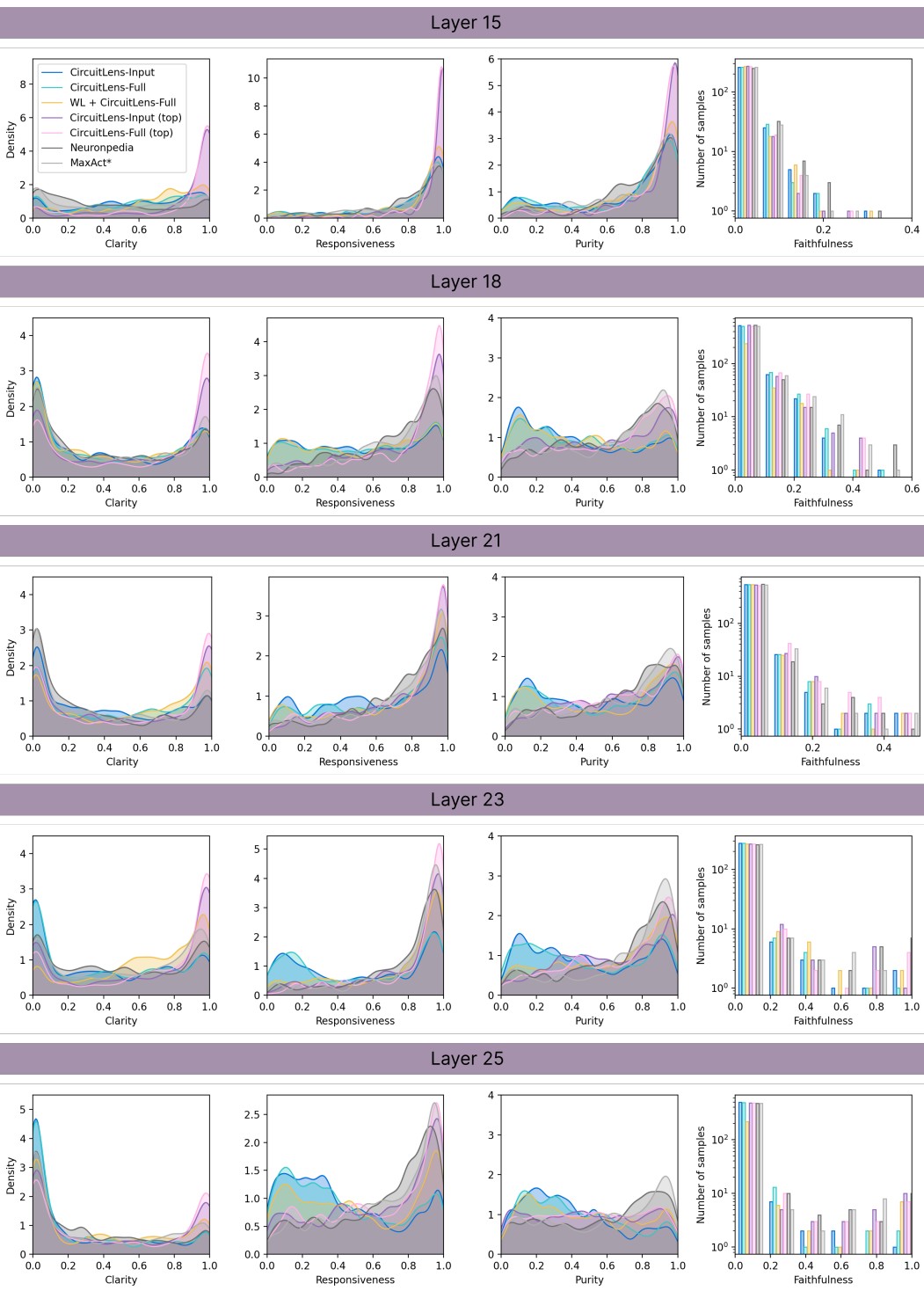

Figure 21: Kernel density estimates illustrating evaluation results of CircuitLens methods in comparison to the baselines (layers 15–25).

