# OpenReview forum: "Circuit Insights: Towards Interpretability Beyond Activations"
_ICLR.cc/2026/Conference — ICLR 2026 Poster_

### Official Review · Reviewer_pB71 · 2025-10-22

**Soundness:** 3
**Presentation:** 2
**Contribution:** 3
**Rating:** 4
**Confidence:** 4

**Summary:**

This paper introduces two new (families of) transcoder feature interpretation techniques. WeightLens techniques rely solely on the feature weights, and their relations to the un/embedding matrices, while CircuitLens techniques go beyond simple maximum feature activations visualizations, to find tokens that are genuinely to the feature as part of a circuit. The authors measure the quality of the interpretations produced by these methods with respect to existing interpretations / methods, and find that their methods perform better.

**Strengths:**

That feature interpretation is a time-consuming and manual process is a well-known issue, and attempts to solve it via auto-interp have fallen short in various ways. This paper proposes a method (or set thereof) that seems better than existing methods; by studying the weights, and features' roles in circuits, it manages to provide more useful information for autointerp models, and produce better interpretations. In this respect, I think the techniques are a novel combination of existing ones, and a significant contribution to the sparse feature interpretation literature, at least in the transcoder context. The authors perform reasonable measurements of interpretation quality, although measuring this is an ongoing and open problem, as I see it.

**Weaknesses:**

**Clarity / Organizational Issues**: It is at times hard to understand which methods are which in this paper. The authors go into great detail re: input-invariant and circuit-based analyses in 3.1 and 3.2, but it's not always obvious how these match up to the methods evaluated in Section 4. You should probably discuss / name the methods in Section 4 in the preceding sections.

Relatedly, it's not always clear which techniques are new to this paper, and which are not. It's clear that generating feature descriptions based on max activations is not new. I also think that "3. Analyze output effects" is a fairly standard application of the logit lens to feature output weights, and also not new. On the other hand, step 1 of the same procedure seems new, as does most of 3.2.

More qualitative examples could help make this paper's methods clearer. I'd love to see examples of the different inputs to the annotator LLM, as well as the resulting labels produced by each method for particular examples. This is important, because it's hard to understand why / how your methods are better without knowing what each method takes in and returns.

In general, more editing of this paper would help. Sentences are often long (233-236), which makes reading hard. I don't think the the bolded-\paragraph-takeaways format works very well for this paper. This is in part because the paper doesn't really telegraph well what claims it's going to make / study. I went into each section not knowing what the authors intended to prove / show, and encountered a collection of findings without a clear narrative to guide them. Similarly, in the methods sections, methods are introduced without it being clear what they would later be used for. This made for a rough reading experience.

**Metrics and Comparisons**: This paper describes the metrics introduced by Puri et al only briefly, while a longer description (with more details about how they are concretely measured) would be appropriate. These metrics are, by my understanding, all powered by LLM-as-a-judge. This is not unreasonable: it's not as though humans can realistically annotate all of these feature descriptions without great effort. However, I think that some sort of control in order to ensure the validity of the LLM judgments (say, a comparison to human annotations) is warranted. More discussion is warranted in general on this front: the challenge of producing better explanations is not just producing good explanations, but also measuring how good they are, which is quite challenging.

Re: comparisons, the authors often compare to Neuronpedia and MaxAct*, but really need to include more details on what these are. How are Neuronpedia's features generated, exactly?

**Usefulness / Scalability**: The authors propose methods for interpreting transcoder features, but right now, there are few transcoders out there. It would be helpful to know if this technique works on other sparse dictionaries, like SAEs. I would also love details on the cost of these techniques (specifically the circuit-based analyses), and if there is an implementation available (as they seem pretty non-trivial to implement).

Despite these weaknesses, I think this paper is near-worth-accepting, and could be convinced to raise my score. I'd love to see the authors improve the clarity of this paper, which I believe will hinder its usefulness for people who are not core mech interp researchers. I also worry that techniques proposed here are a bit too niche and expensive to be practically useful / interesting for a broader audience, and have asked questions about this below.

**Questions:**

- Can this method be used with SAEs (as opposed to transcoders)?
- What are the computational requirements of each proposed method?
- Why do you (in 3.1) also consider contributions of tokens via earlier features? This kind of seems like a manual, 1-step version of attribution (which tries to avoid the fact that you don't actually have activations for your transcoders, since you're not running them on an input). It seems more natural to me to just use direct effects, or to actually do attribution.
- You seem to believe that sampling from various quantiles makes more sense for sparse dictionaries than just interpreting max acts (which is better for neurons). Do you find any empirical evidence for this?
- I find 3.2 rather hard to read, as its framing is a bit strange. Why do you say that "To account for interference between layers, Ameisen et al. (2025) propose incorporating a Jacobian term into the attribution formulation..."? I don't think they're accounting for interference, per se; they're just computing the direct effects of one feature on another via all possible paths, which can be done using a Jacobian term. More pertinently, you haven't really introduced attribution yet, so this section comes a bit out of the blue.
- What exactly are the inputs you use for the circuit-based methods, which require inputs with respect to which to perform attribution? The authors say "For generating circuit-based descriptions, we use sampling, described in the Subsection 3.2", but I think it'd be good to be clear about the whole pipeline. I think that what happens is that they sample sentences using inverse quantile frequency, then perform attribution on those samples, and then use these to generate descriptions.
- Can you make Figure 4 easier to read?

---

> ### Author Response · Authors · 2025-11-21
> **Response to Reviewer pB71 (1/3)**
>
> We thank the reviewer for highlighting the significance and novelty of our work. With the answers provided here and the revisions in our updated work, we hope we have addressed the weaknesses and questions raised.
> ___
>
> **[W1 - Clarity / Organizational issues]**
>
> **[Methods not clearly linked to experiments]** The methods we evaluate in Section 4 are concrete instantiations of the methodology described in Section 3. In the revised version we describe these setups clearer so that they are better connected to Section 3, as well as revising methodology part for clarity, making it easier to understand what comes from which part of the methodology. We hope this helps the reader better understand how the methodology translates into the evaluated experiments.
>
> **[Novel vs. prior contributions unclear]** Analyzing output effects via unembedding projection (not the logit lens) for automated interpretability is indeed not a contribution of our work, and we have now added an explicit citation to the prior paper that introduced this approach. We reorganized Section 3 to provide a clearer separation between our original contributions and existing work.
>
> **[More qualitative examples needed]** Figures 5 and 6 illustrate the difference between the inputs provided to the explainer LLM in the baselines, namely raw activations, and the inputs we provide after identifying patterns that drive the feature’s activation. Additional examples have been added to Appendix E due to page limits, where we demonstrate explicitly the differences in the inputs, provided to the explainer LLM for baselines and for CircuitLens.
>
> **[Clarity and narrative flow]** We have improved the organization and motivation of Section 3 and refined the overall formatting to make the paper smoother and easier to read. Sections 4 and 5 have also been revised to clarify the motivation behind the experiments and results discussion, ensuring a more coherent narrative throughout the paper.
> ___
>
> **[W2 - Metrics and Comparisons]**
>
> We expanded the metric descriptions in the main text and added details in the Appendix A.2. Methods aggregated in FADE framework [1] are widely used across automated interpretability research, and earlier work shows they correlate well with human judgments of description quality. This supports their validity as LLM-as-a-judge evaluations, although we agree that further comparisons to human annotations would strengthen future work. We now include additional qualitative examples and will release all outputs and the full codebase to support reproducibility and community verification.
>
> Details on the baselines are now included in Appendix D. Neuronpedia descriptions are based on Bills et al. [2], while MaxAct\* follows the FADE evaluation procedures [1]. These methods extract top-activating examples from a dataset and pass them to an explainer LLM, either highlighting activated tokens (e.g., <<\<this\>>>) or providing [token, activation] pairs.

---

> ### Author Response · Authors · 2025-11-21
> **Response to Reviewer pB71 (2/3)**
>
> **[W3 - Usefulness / Scalability]**
>
> **[Transferability to other architectures]** WeightLens is specific to transcoders, since their architecture enables a clean separation between input-dependent and input-invariant components in the attribution signal. By contrast, CircuitLens is not explicitly tied to transcoders. It can be extended to SAEs or other sparse dictionaries using gradient-based attribution methods. While this is out of the scope of this work, it would be interesting to explore this aspect in the future work.
>
> **[Computational costs]** CircuitLens input-based analysis requires one forward and one backward pass per input, resulting in ~18 seconds per feature when using 100 sampled inputs (5 hours for 1000 features on an A100 40GB GPU). Output-based analysis performs an additional forward-backward pass for each generated token. With 15 generated tokens per input and the same 100 inputs, this amounts to ~45 seconds per feature (12 hours for 1000 features).
>
> However, using 100 inputs per feature and 15 generated tokens per input is intentionally conservative and larger than what is typically needed. Most automated interpretability works use 5-20 inputs per feature; we selected 100 to reliably capture all possible activation clusters. However, this number could likely be reduced with minimal loss in quality, although further experiments would be required to determine the point at which quality begins to degrade.
>
> Finally, CircuitLens relies on a much smaller dataset than MaxAct* and other activation-based baselines, which further reduces the compute and memory needed for activation caching.
>
> **[Implementation availability]** The full implementation is included in the supplementary material, and we will provide a public GitHub link in the camera-ready version.
> ___
>
> **[Q1 - Extension to SAEs]** As mentioned in [W3], WeightLens is specific to transcoders due to their architecture (see Dunefsky et al. [4]). However, the core ideas behind CircuitLens are general and can be adapted to other architectures.
>
> **[Q2 - Computational costs]** WeightLens is lightweight and runs locally on MacBook Pro M3 Pro via MPS. For Gemma-2-2b  ~0.04 seconds per feature for early layers (layer 0) and ~0.35 seconds per feature at layer 25, increasing linearly through layers. For CircuitLens computational costs are higher; details are presented in Appendix C and reported in this answer under [W3].
>
> **[Q3 - Token contributions via earlier-layer features]** We observe that many strong connections between features on a given input are also reflected in weight-based connections. For some features, the direct effect of embeddings through WeightLens is not reflecting the feature’s activation, the strongest connections to the vocabulary may appear random or misleading. However, when considering the input-invariant part of the attribution (i.e. weights) from feature to feature, there are many meaningful connections. In such cases, propagating descriptions from earlier-layer features allows us to describe the current feature meaningfully even when direct effects are weak. This approach builds explanations layer by layer, leveraging existing descriptions from previous features. We now discuss this thoroughly in Appendix E and provide additional qualitative examples.
>
> **[Q4 - Sampling over the whole distribution]** Max-activation sampling sometimes captures outliers rather than the general behavior of a feature [3]. Many autointerpretability studies sample from the top percentile of activation.  However, such sampling often fails to capture the full distribution of a feature’s activations and behaviors:
>
> 1. Polysemantic features exhibit multiple distinct activation patterns corresponding to different meanings. For example, a Gemma-2-2b feature activates on matplotlib plot commands, the word “receptor,” and references to “Fig.” or “Figure” in academic contexts, but the description is only “snippets of python code using matplotlib”
> (https://www.neuronpedia.org/gemma-2-2b/3-gemmascope-transcoder-16k/9607).
>
> 2. Top-activation sampling can also yield overly narrow descriptions. A feature that strongly activates on “ethics” or “ethics committees” also activates on councils, foundations, committees, and other organizations in research or medical contexts, often in expressions of gratitude or acknowledgment (https://www.neuronpedia.org/gemma-2-2b/21-gemmascope-transcoder-16k/40). Focusing only on the highest activations obscures these broader patterns.

---

> ### Author Response · Authors · 2025-11-21
> **Response to Reviewer pB71 (3/3)**
>
> **[Q5 - Attribution Question]** We build on the work of Dunefsky et al. [4], which introduces feature-to-feature attribution mentioned at the beginning of Section 3. In their formulation, nonlinearities between layers are not accounted for, and no Jacobian term is included. This works well for GPT-2, but when transferring experiments to Gemma and Llama models, we observed that the same approach does not perform as effectively. The attribution formulation proposed by Ameisen et al [5]. introduce a Jacobian term, which we interpret as a way to account for higher nonlinearity in modern models that use RoPE and exhibit slightly different interpretability behavior compared to GPT-2. We now gather all attribution details in Section 3.1, making the methodology more intuitive and connected.
>
> **[Q6 - Circuit-based input pipeline]** The reviewer correctly describes the pipeline.  We first cache feature activations over a dataset of 24M tokens, saving the maximum activation per sample. We then perform inverse quantile sampling, ensuring we capture general feature behavior rather than only outliers. For each feature, we select 100 samples and compute attribution with respect to the maximally activating token. From these attributions, we:
>
> 1. Identify input patterns that led to feature activation on the target token and mask them for future description generation. We also allow the model to generate new tokens after the maximally activating token, and analyze the feature’s contribution to each generated token.
>
> 2. Cluster based on circuits identified to participate in feature’s activations to obtain monosemantic, interpretable clusters.
>
> 3. Use patterns from each cluster to generate cluster-specific descriptions, which are then combined into a single feature description.
>
> We reorganized Section 3.2 (now Section 3.3) to provide a more intuitive and stepwise overview of the methodology.
>
> **[Q7 - Figure 4 readability]** We made Figure 4 larger for improved readability.
> ___
>
> We thank the reviewer for their detailed feedback and thoughtful suggestions. We hope that the clarifications provided here, together with the reorganized Sections 3, 4 and 5, have addressed the concerns regarding presentation and clarity, as well as used metrics and computational costs. We believe these revisions make the work stronger and more accessible, and we kindly ask the reviewer to consider increasing their score. We are happy to provide further clarifications or engage in additional discussion if helpful.
> ___
> **References**
>
> [1] Puri et al., FADE: Why Bad Descriptions Happen to Good Features, ACL Findings 2025.
>
> [2] Bills et al., Language models can explain neurons in language models, 2023.
>
> [3] Bykov et al., DORA: Exploring Outlier Representations in Deep Neural Networks, Transactions on Machine Learning Research (TMLR), 2023.
>
> [4] Dunefsky, J., Chlenski, P., Nanda, N., Transcoders Find Interpretable LLM Feature Circuits, arXiv:2406.11944, 2024.
>
> [5] Ameisen, E., Lindsey, J., Pearce, A., et al., Circuit Tracing: Revealing Computational Graphs in Language Models, Transformer Circuits Thread, 2025.

---

> > ### Comment · Reviewer_pB71 · 2025-11-24
> >
> > > [Novel vs. prior contributions unclear] Analyzing output effects via unembedding projection (not the logit lens) for automated interpretability is indeed not a contribution of our work, and we have now added an explicit citation to the prior paper that introduced this approach. We reorganized Section 3 to provide a clearer separation between our original contributions and existing work.
> >
> > Do you not call this the logit lens because you're applying the unembedding matrix to the output weights rather than activations? Or is there a substantial difference here?
> >
> > > The attribution formulation proposed by Ameisen et al [5]. introduce a Jacobian term, which we interpret as a way to account for higher nonlinearity in modern models that use RoPE and exhibit slightly different interpretability behavior compared to GPT-2. We now gather all attribution details in Section 3.1, making the methodology more intuitive and connected.
> >
> > Thanks for putting them all in one place! That helps. But that is not a correct interpretation of the Jacobian term. Ameisen et al. account for nonlinearities by detaching them (typically the attention patterns, layernorm denominators, and the features themselves). The Jacobian term simply traces the many paths from one feature input to the other feature's output.
> >
> > > [Q7 - Figure 4 readability] We made Figure 4 larger for improved readability.
> >
> > Sorry, I should have been clearer - it's the choice of different bar fills + the narrowness of the bars that makes this one hard to read. You should probably distinguish by metric using color, and use only two different fills; this will be much easier to read, at least in color.
> >
> > Also, there's a new typo in the title of subsection 3.3.
> >
> > ---
> >
> > Overall, I appreciate the work that the authors have done to improve the readability of this paper. I still find it rather challenging to work through, even as someone who is pretty familiar with this topic; I get the sense that this is in part due to how complicated the methods are. I feel like this is mostly good work, but find myself a bit hesitant to recommend it because of its complexity and computational cost (at least on the CircuitLens side).
> >
> > I wonder if e.g. some of the benefits of CircuitLens with respect to finding multiple clusters of activations could have been attained by performing clustering on a different quantity (after all, I don't think any other methods even consider polysemanticity). Similarly, could the authors have constructed a stronger baseline combining only the simplest improvements brought by WeightLens - e.g. run an LLM with access to activations as well as the top / bottom output logits / input tokens? I think this method works (according to the metrics the authors use), but I'm not 100% sure of which parts are necessary, and if the extra complication is worth it.

---

> > > ### Author Response · Authors · 2025-11-26
> > > **Response to Reviewer pB71**
> > >
> > > We thank the reviewer for clarifying the concerns about Figure 4 and for pointing out the typo. We will review the work again and update Figure 4 for improved visibility in the revised version.
> > >
> > > >Do you not call this the logit lens because you're applying the unembedding matrix to the output weights rather than activations? Or is there a substantial difference here?
> > >
> > > Exactly. The reviewer’s interpretation is correct: logit lens projects residual stream activations through the unembedding matrix [1], while WeightLens applies the projection to a feature’s decoder vector. The mechanism is similar, but WeightLens shows which tokens the feature would promote or suppress. We avoided calling it logit lens to prevent confusion.
> > >
> > > >“Ameisen et al. account for nonlinearities by detaching them (typically the attention patterns, layernorm denominators, and the features themselves). The Jacobian term simply traces the many paths from one feature input to the other feature's output.”
> > >
> > > We agree with the reviewer. Our earlier wording may have implied a stronger claim than intended and reflected our interpretation rather than the original work. Our goal was to highlight why we use the attribution formulation by Ameisen et al. [2] (Eq. 2) rather than Dunefsky et al. [3] (Eq. 1).
> > >
> > > Dunefsky et al. track direct feature influence in GPT-2 via forward-pass activations, decoder, and encoder weights, capturing meaningful direct contributions. Experimentally we found that applying the same method to Gemma-2-2B and Llama-3.2-1B transcoders gave unsatisfactory results, even for adjacent layers. This suggests that direct contributions alone may not be sufficient in these models, and that indirect influences via attention heads play an important role. One possible factor is that these models use RoPE, which introduces position-dependent interactions in attention and can make some feature influences less direct and not fully captured by forward-pass-only attribution. Computing the Jacobian with frozen nonlinearities allows these indirect effects to be incorporated.
> > >
> > > We will reframe this section to prevent further misunderstanding.
> > >
> > > >I feel like this is mostly good work, but find myself a bit hesitant to recommend it because of its complexity and computational cost (at least on the CircuitLens side). I wonder if e.g. some of the benefits of CircuitLens with respect to finding multiple clusters of activations could have been attained by performing clustering on a different quantity (after all, I don't think any other methods even consider polysemanticity).
> > >
> > > We appreciate the reviewer's positive assessment. While feature-level analysis is more expensive than activation-only methods due to the backward pass, overall compute can still be moderate: we achieve comparable-quality results with a much smaller underlying dataset.
> > >
> > > This analysis also provides benefits that activation-only methods cannot offer:
> > >
> > > 1) it shows which specific parts of the input caused the feature to activate, often not reflected in the feature’s activations;
> > >
> > > 2) it shows how a feature influences the model’s output by identifying which token predictions it contributes to;
> > >
> > > 2) it enables reliable clustering based on underlying circuits, including non-semantic features such as syntactic or structural patterns.
> > >
> > > Clustering has been explored before and shown to improve description quality for SAEs [4]. However, prior work applied it to embeddings, which are mainly suited for separating semantic concepts. Many features instead activate on structural or positional patterns, where embedding-based clustering is less meaningful. Our approach of circuit-based clustering is a novel and effective way to disentangle these behaviors at different granularities, as shown in Appendix E.
> > >
> > > >Similarly, could the authors have constructed a stronger baseline combining only the simplest improvements brought by WeightLens - e.g. run an LLM with access to activations as well as the top / bottom output logits / input tokens?
> > >
> > > We thank the reviewer for the suggestion. We agree that such a baseline would be a useful addition to isolate the contribution of WeightLens. Due to time constraints during the discussion phase, we are unlikely to run it immediately, but we plan to include it in the camera-ready version.
> > >
> > > **References**
> > >
> > > [1] nostalgebraist. (2020, August 31). Interpreting GPT: the logit lens. LessWrong. https://www.lesswrong.com/posts/AcKRB8wDpdaN6v6ru/interpreting-gpt-the-logit-lens
> > >
> > > [2] Ameisen, et al., "Circuit Tracing: Revealing Computational Graphs in Language Models", Transformer Circuits, 2025.
> > >
> > > [3] Dunefsky, J., Chlenski, P., & Nanda, N. (2024). Transcoders find interpretable llm feature circuits. Advances in Neural Information Processing Systems, 37, 24375-24410.
> > >
> > > [4] Kopf, L., Feldhus, N., Bykov, K., Bommer, P. L., Hedström, A., Höhne, M. M. C., & Eberle, O. (2025). Capturing Polysemanticity with PRISM: A Multi-Concept Feature Description Framework. arXiv preprint arXiv:2506.15538.

---

### Official Review · Reviewer_gvCa · 2025-10-25

**Soundness:** 2
**Presentation:** 2
**Contribution:** 3
**Rating:** 4
**Confidence:** 4

**Summary:**

This paper introduces WeightLens and CircuitLens, two methods for automatically generating feature descriptions. WeightLens analyzes model weights alone to identify token-level features and their stable, input-invariant connections, while CircuitLens traces dynamic, input-dependent interactions between features, attention heads, and output tokens using Jacobian-based attributions. By clustering using Jaccard similarity and DBSCAN, the approach tries to generate multiple descriptions of a feature to incorporate its polysemantic behavior.

**Strengths:**

1. This work addresses an important problem of generating automatic feature descriptions without relying on black-box LLMs or large-scale datasets.
2. The proposed WeightLens technique offers a practical solution to these limitations by producing competitive, and in some cases superior, feature descriptions compared to activation-based methods, while remaining relatively simple to implement.
3. Moreover, the integration of both input- and output-centric analyses, combined with the use of Jaccard similarity and DBSCAN for clustering input examples based on circuit components, is a particularly interesting idea.

**Weaknesses:**

1. The main motivation of this work is to eliminate dependence on black-box explainer LLMs and large datasets. However, the CircuitLens method still requires access to both a secondary LLM and a large dataset. Therefore, only the WeightLens method truly addresses the core problem the paper aims to solve.
2. Figure 1 is not very informative in its current form. It could be significantly improved by expanding it into a larger figure composed of subfigures illustrating each of the three steps. For example, consider a random feature f at layer l, and visually depict all the steps required to obtain its token-level descriptions.
3. It would also help readers if the Circuit-based analysis subsection (Section 3.2) were better contextualized in relation to the previous subsection and the overall goal of the work. You could begin the section with a brief paragraph explaining why circuit-based analysis is necessary to obtain meaningful feature descriptions.
4. Section 3.2 is somewhat difficult to follow. Overall, the process for generating feature descriptions involves the following steps:
   1. Sample examples with varied activation values using the proposed sampling strategy.
   2. For each example, perform input- and (optionally) output-centric analyses.
   3. Compute the Jaccard similarity matrix.
   4. Apply DBSCAN to cluster the input examples.
   5. Use an LLM to generate cluster-level and overall feature descriptions.

   However, this workflow is not clearly conveyed in the current text, particularly due to the ordering of explanations. I recommend reorganizing this section to follow the logical flow outlined above.

5. The evaluation results for both WeightLens and CircuitLens are presented for only four layers. Results across all (or most) layers should be included, preferably in the appendix, to enable a more comprehensive assessment of their effectiveness.
6. (Minor point) It would be easier to compare different configurations of both WeightLens and CircuitLens if all the evaluation criteria could be aggregated into a single one.

**Questions:**

1. It is mentioned that a z-score is used to identify outlier tokens. What threshold value is applied to filter these tokens?
2. In Step 2 (Validate tokens) of Section 3.1, tokens that actually activate the feature are filtered in. What criterion defines a token as activating a feature, i.e. what activation threshold is used to make this determination?
3. In WeightLens, it is unclear how the lemmatized tokens are transformed into feature descriptions. Is an LLM employed for this step?
4. Can you also clarify whether output-centric analysis is used when computing the Jaccard similarity? The paragraph describing circuit-based clustering mentions collecting transcoder features and token/attention head pairs for each input but does not specify whether the feature’s impact on the final logit is considered.
5. Lastly, do the activation-based baseline methods use the same secondary LLM to generate feature descriptions?

---

> ### Author Response · Authors · 2025-11-21
> **Response to Reviewer gvCa (1/2)**
>
> We thank the reviewer for highlighting the importance of automated feature interpretation and the practical contribution of WeightLens. We are glad that our combination of input- and output-centric analyses with circuit-based clustering, as well as the use of weight- and circuit-based information to improve sparse feature interpretation, was appreciated. Below, we address the weaknesses and questions the reviewer mentioned.
> ___
>
> **[W1 - CircuitLens depends on explainer LLM and a dataset]** We acknowledge that fully eliminating reliance on datasets or an explainer LLM remains challenging for complex, context-dependent features. To clarify this point, we have slightly revised the contributions paragraphs and explicitly highlighted it in the limitations.
>
> With CircuitLens our goal is to reduce reliance on the explainer LLM capabilities by simplifying its task: we provide specific input patterns that trigger a feature, rather than requiring the LLM to identify these patterns on its own. At the same time, we reduce reliance on large datasets, especially when combining weight and circuit-based information, allowing comparable results on a dataset of 24M tokens, compared to 3.6B tokens used by the MaxAct* baseline.
>
> **[W2 - Figure 1 change suggestion]** We have revised Figure 1 (now it is Figure 2)  to clearly illustrate the process of generating token-level feature descriptions, and we clarified Section 3.1 to make the algorithm and its motivation easier to follow.
>
> **[W3 and W4 - Organization of Section 3.2]** We thank the reviewer for the helpful suggestion. We have reorganized Section 3 to decouple the attribution calculation equations from the CircuitLens method itself to make it cleaner. Section 3.2 (3.3 in the revised version) now follows the actual workflow for interpreting features (activations caching and sampling → input/output analyses → clustering → LLM descriptions) and includes additional context and motivation for circuit-based analysis. We hope these revisions make the section clearer and more intuitive for readers.
>
> **[W5 - Experiment on limited number of layers]** We selected layers [0, 7, 12, 21] by uniformly sampling across the transformer to balance computational cost while capturing the full spectrum of representations. Additional layers are being evaluated, and more results will be included before the end of the discussion period. Some experiments, particularly MaxAct* and CircuitLens (top), are highly computationally intensive and will likely only be ready for the camera-ready version.
>
> We focus on a limited number of layers because reliable conclusions require processing hundreds of features per layer due to high variance in features interpretability level. Prior work also often uses a limited set of layers [1,2,3], while others reporting more layers either rely on inexpensive metrics [4] or evaluate only a small number of features per layer (e.g., 40 [5]), which we find insufficient. This motivates our evaluation of hundreds of features per layer (e.g., 500 for CircuitLens) while restricting detailed analysis to a subset of layers. We hope this clarification, along with the additional experimental results that we aim to provide as soon as possible, addresses this point.
>
> **[W6 - Aggregate FADE metrics]** We appreciate the reviewer’s suggestion regarding aggregation of the evaluation criteria. We use the FADE framework precisely because it captures multiple complementary aspects of description quality, highlighting the strengths and limitations of each method. While aggregating these metrics into a single score could simplify comparisons, it may also obscure important distinctions that are crucial for interpreting the results. We therefore chose to present the metrics separately, which we believe provides a more informative and nuanced discussion.

---

> ### Author Response · Authors · 2025-11-21
> **Response to Reviewer gvCa (2/2)**
>
> **[Q1 - Z-score threshold]** The z-score threshold for identifying outlier tokens depends on the model and the transcoders and was adjusted qualitatively. For embeddings outliers, we used a threshold of 4 for GPT-2 and 4.5 for Gemma-2-2B and LLaMA-3.2-1B, reflecting their larger vocabularies and the need to select only the most extreme tokens. For identifying connected features, we use a threshold of 3. This information is now reflected in Section 3.
>
> **[Q2 - Activation threshold for validation step]** The decision is binary: a token either activates the feature or it does not (activation is 0). Since activations are not cached at this step, we cannot determine the distribution of activation values or a meaningful threshold. For sparse activations, we assume that any non-zero activation is sufficient to include the token.
>
> **[Q3 - Lemmatization for WeightLens descriptions]** No LLM is used at the stage of WeightLens postprocessing (only in a separate experiment where it is explicitly mentioned). We apply the WordNetLemmatizer from NLTK to all identified tokens (e.g., ['all', '_all', 'All', 'ALL', '_All', 'tout']) and then take the set of unique lemmatized forms (e.g., ['all', 'tout']). These results are then used to construct the feature description.
>
> **[Q4 - Input/Output-centric]** Output-centric analysis is not used when computing the Jaccard similarity or performing clustering. In preliminary experiments we considered combining input- and output-based clustering, but output-based clustering was often redundant or noisy. Empirically, we observe that a feature’s contribution to the output is largely determined by the activations of earlier-layer features that drive it. That is, while downstream interactions can occur, the feature’s effect on the output is already largely encoded by the upstream circuits responsible for its activation. Therefore, clustering based on these input-driven patterns is sufficient for our task.
>
> **[Q5 - Explainer LLM]** We use GPT-4o-mini as the main model for generating feature descriptions and performing evaluations, including for the MaxAct\* baseline. In contrast, the Neuronpedia baseline descriptions were generated using Gemini 2.0 Flash, which is generally stronger than GPT-4o-mini.
> ___
>
> We thank the reviewer for their thoughtful feedback and hope that our clarifications, together with the revisions adequately address the concerns and weaknesses highlighted. In particular, we hope the explanations regarding layers selection [W5], reorganization of CircuitLens workflow section [W3/W4], and broader methodological clarifications in response to the reviewer’s questions provide a clearer understanding of our approach. If the reviewer finds that these clarifications sufficiently resolve their concerns, we would kindly ask them to consider increasing their score. We remain happy to provide further explanations if needed.
> ___
>
> **References**
>
> [1] Kopf et al., Capturing Polysemanticity with PRISM: A Multi-Concept Feature Description Framework, arXiv:2506.15538, 2025.
>
> [2] Lubana et al., Priors in Time: Missing Inductive Biases for Language Model Interpretability, arXiv:2511.01836, 2025.
>
> [3] Puri et al., FADE: Why Bad Descriptions Happen to Good Features, ACL Findings 2025.
>
> [4] Choi et al., Scaling automatic neuron description, 2024.
>
> [5] Gur‑Arieh et al., Enhancing Automated Interpretability with Output‑Centric Feature Descriptions, ACL 2025.

---

> > ### Comment · Reviewer_gvCa · 2025-11-24
> >
> > Thank you for the detailed and thoughtful rebuttal. After reviewing the revised manuscript and the clarifications provided, I believe that several of my earlier concerns have been meaningfully addressed. While some limitations remain, especially regarding full independence from datasets and explainer LLMs, the authors acknowledge these constraints transparently and appropriately adjust the framing of their contributions. Hence, I’m increasing my score.

---

### Official Review · Reviewer_ZcNR · 2025-10-29

**Soundness:** 3
**Presentation:** 2
**Contribution:** 3
**Rating:** 6
**Confidence:** 3

**Summary:**

This paper proposes WeightLens and CircuitLens as two complementary methods for sparse feature interpretation. WeightLens leverages the input-invariant term of the transcoder attribution to explain context-independent features, while CircuitLens use attribution scores to capture how feature activations arise from interactions between components. These methods show higher clarity and responsiveness than traditional activation-based interpretation.

**Strengths:**

1. Current methods for automated interpretation of SAE features largely depend on observing the feature activation pattern, which is restricted in explaining complex features with deep connection with upstream and downstream features. This paper takes advantage of transcoders and attribution scores to extend the information for interpreting features. It follows a natural direction of the progress of SAE feature interpretation.
2. The sampling strategy in Section 3.2 considers the complete spectrum of feature activation magnitudes, providing comprehensive explanation of the feature's behavior.
3. The WeightLens method eliminates the need for an explainer LLM.
4. The interpretability metrics from the FADE framework gets substantial improvement with CircuitLens.

**Weaknesses:**

1. The motivation of several parts of the paper remains opaque and the expression is unclear, including Assumption 2 in Section 3.1, Circuit-Based Clustering in Section 3.2. This makes the paper hard to follow.
2. The faithfulness score seems very low using either WeightLens or CircuitLens. Whether the interpretation truly reflects the features' behavior is doubtful.

**Questions:**

1. In generating feature description in Section 3.1, is the connection between transcoders also results in token descriptions? Why is this necessary instead of directly showing inter-feature behavior in description?
2. Can you provide more example of the results of WeightLens & CircuitLens?

---

> ### Author Response · Authors · 2025-11-21
> **Response to Reviewer ZcNR**
>
> We thank the reviewer for recognizing the value of incorporating circuit-based information for automated feature interpretability, as well as the benefits of our sampling strategy and the LLM-free design of WeightLens. With the clarifications and revisions outlined below, we aim to fully address the reviewer’s concerns and strengthen the presentation of our contributions.
> ___
>
> **[W1 Opaque Motivation]** We thank the reviewer for pointing out that the motivation for Assumption 2 and circuit-based clustering was unclear. We have revised the Methodology section for clarity in the updated manuscript. The motivation for Assumption 2 is now explicitly stated as a consistency check between token-based feature descriptions and observed behavior, and we clarify why circuit-based clustering is needed for features involving more complex syntactic or logical patterns.
>
> **[W2 Low Faithfulness Scores]** We agree with the reviewer, that the demonstrated scores of faithfulness are low. Most implementations of the Faithfulness score are designed for residual stream SAEs, where ablating or steering features works well, especially in later layers. In transcoders, which write into the residual stream like an MLP layer, such interventions are less impactful due to high redundancy of features and distributed representation: similar behaviors often appear across multiple layers, and changing a single feature’s activation is rarely having strong influence on the result. Prior work has demonstrated downstream effects by manipulating clusters of related features, though those clusters were manually defined for each circuit [1]. A natural future direction would be to measure faithfulness at the level of the identified circuit.
> Thus, *this is an architectural property, not specific to our method, and affects baselines as well, which we now specifically mention in limitations.*
> ___
>
> **[Q1 - Transcoder connections for token descriptions]** In Section 3.1, the feature descriptions are always token-based, not feature-to-feature. We include connections to earlier-layer features because, for many features, the direct connection to vocabulary tokens is weak or non-informative, while the feature-to-feature connections remain meaningful. In such cases, tokens that reliably describe an upstream feature can also help describe the current feature. This allows us to propagate token-level information not only directly through embeddings but also indirectly through the network of features, capturing patterns that would otherwise be missed. Thus, we are not using the connections themselves as a description; rather, we propagate token-level contributions along these meaningful feature-to-feature paths. We now clarify this mechanism further and provide additional examples in the Appendix E.
>
> **[Q2 - Examples Request]** We have added a new section in the appendix (Appendix E) containing detailed case studies that describe the motivation behind our methods. We demonstrate several examples on WeightLens, showing how embeddings- and feature-based connections are resulting in a feature’s description, as well as provide detailed examples on CircuitLens inputs masking and circuit-based clustering. We hope that these additions further strengthen the presentation of our work and its contributions.
> ___
>
> We thank the reviewer for their detailed feedback and hope that our clarifications, along with the revisions and additional examples provided in the appendix, address their concerns and improve the presentation of our work. We would be happy to engage further and answer any remaining questions. If the reviewer finds that these changes satisfactorily resolve their points, we would kindly ask them to consider raising their score.
> ___
>
> **References**
>
> [1] Lindsey, et al., On the Biology of a Large Language Model, Transformer Circuits, 2025.

---

> > ### Comment · Reviewer_ZcNR · 2025-11-24
> >
> > Thanks for the detailed responses from the authors. Given the current writing quality and the contribution of the paper, I'll keep my score.

---

### Author Response · Authors · 2025-12-03
**Discussion Summary and Remarks**

We appreciate the constructive rebuttal period and the thoughtful engagement from all reviewers. We thank reviewer gvCa for suggestions that led to concrete improvements and for subsequently raising their score, reviewer ZcNR for the positive assessment and helpful feedback, and reviewer pB71 for a detailed discussion that helped us refine the narrative and methodology. We regret that the discussion period was cut short and we were unable to receive the reviewer’s final feedback, though we hope our responses addressed their concerns.

During the rebuttal, we focused on improving clarity, organization, and completeness. The main updates to the manuscript include:

- Clarified methodology and motivation, including Assumption 2 and circuit-based clustering.

- Reorganized Sections 3.1-3.3 for clearer workflow and connections to the evaluated experiments.

- Updated Figures 2 and 4 and refined the organization and flow of Sections 4 and 5 to improve clarity of the results.

- Added detailed examples in Appendix E showing WeightLens and CircuitLens outputs.

- Extended experimental results to additional layers in Appendix G, addressing reviewer gvCa’s request.

- Explicitly stated limitations and computational costs.

These improvements strengthen the overall clarity and coherence of the paper while preserving the aspects highlighted by reviewers as particularly valuable. In particular, WeightLens offers fast, dataset- and model-independent automated feature interpretation, CircuitLens provides a novel way to analyze circuits and disentangle complex feature behaviors, and both methods produce high-quality, informative feature descriptions that compare favorably to activation-based baselines.

We hope that these updates, together with the contributions recognized by the reviewers and the increased scores after the rebuttal (6/6/4, with the third reviewer not able to provide a final rating), make the work sufficiently clear and impactful for acceptance.

---

### Meta-Review · Area_Chair_kFFv · 2025-12-12

**Summary:**

This study proposes two methods that aim to reduce the data and labor requirements for current interpretability studies. The first is WeightLens, an approach that interprets the weights of features with respect to the (un)embedding matrices. The second is CircuitLens, a method based on gradient attributions that takes feature interactions into account.

Reviewers agree that these methods appear to be genuine improvements over prior methods in reducing human effort. There was also agreement that the combination of methods employed here is novel. However, reviewers unanimously agree that there are significant clarity issues in both the presentation of the methods and the motivation, and many of these seem to remain after the discussion period. There were also concerns regarding the generality of WeightLens due to reliance on transcoders in particular, which are not common and unlikely to remain the state of the art in dictionary learning. Overall, reviewers agreed that this paper could be of use to those in the interpretability community, but with some reservations.

**Reviewer Concerns:**

The primary concerns shared across reviewers related to clarity. Reviewers ZcNR and gvCa agreed that the general motivation for some methods was unclear. Reviewers gvCa and pB71 agreed that the technical sections were difficult to follow, and that the claims were also unclear in some cases. The authors made a good effort to clarify many of these details, but the reviewers share hesitation on this point even after the discussion given how pervasive this issue seemed to be.

There was also a concern from Reviewer pB71 regarding generality: transcoders are required for WeightLens. This has not been addressed.

The low faithfulness scores were raised by Reviewer ZcNR; this has been addressed. Reviewer gvCa mentioned that CircuitLens still requires significant compute, but the discussion has clarified that this method still requires significantly less than prior methods.

**Reviewer Scores:**

Reviewer ZcNR has stated that they will not change their score. Reviewer gvCa had increased their score originally. Reviewer pB71 did not have a chance to continue the discussion with the authors, but I sense that their reservations were not sufficiently addressed by the authors to warrant a score increase.

---

### Decision · Program_Chairs · 2026-01-26

Accept (Poster)